EMBO
Molecular Medicine

# Epithelial response to IFN-γ promotes SARS-CoV-2 infection

Julian Heuberger[1,2,3], Jakob Trimpert[4], Daria Vladimirova[4], Christian Goosmann[2], Manqiang Lin[1], Rosa Schmuck[5], Hans-Joachim Mollenkopf[2], Volker Brinkmann[2], Frank Tacke[1], Nikolaus Osterrieder[4,6] & Michael Sigal[1,2,3,*] (ID)

## Abstract

SARS-CoV-2, the agent that causes COVID-19, invades epithelial cells, including those of the respiratory and gastrointestinal mucosa, using angiotensin-converting enzyme-2 (ACE2) as a receptor. Subsequent inflammation can promote rapid virus clearance, but severe cases of COVID-19 are characterized by an inefficient immune response that fails to clear the infection. Using primary epithelial organoids from human colon, we explored how the central antiviral mediator IFN-γ, which is elevated in COVID-19, affects epithelial cell differentiation, ACE2 expression, and susceptibility to infection with SARS-CoV-2. In mouse and human colon, ACE2 is mainly expressed by surface enterocytes. Inducing enterocyte differentiation in organoid culture resulted in increased ACE2 production. IFN-γ treatment promoted differentiation into mature KRT20[+] enterocytes expressing high levels of ACE2, increased susceptibility to SARS-CoV-2 infection, and resulted in enhanced virus production in infected cells. Similarly, infection-induced epithelial interferon signaling promoted enterocyte maturation and enhanced ACE2 expression. We here reveal a mechanism by which IFN-γ-driven inflammatory responses induce a vulnerable epithelial state with robust replication of SARS-CoV-2, which may have an impact on disease outcome and virus transmission.

**Keywords** ACE2; differentiation; interferon; organoids; SARS-CoV-2
**Subject Categories** Immunology; Microbiology, Virology & Host Pathogen Interaction

## Introduction

Severe acute respiratory syndrome coronavirus 2 (SARS-CoV-2) causes coronavirus disease (COVID-19) and has already infected more than 80 million people around the globe. Patients develop highly variable clinical symptoms that range from mild to life-threating. SARS-CoV-2 invades epithelial cells of the mucosal surfaces of the upper and lower respiratory tract as well as the gastrointestinal tract, but can also replicate in epithelia of other organs (Zhu et al, 2020). In addition to respiratory symptoms, gastrointestinal symptoms are common in COVID-19 and recent data revealed that the virus can efficiently infect enterocytes (Cholankeril et al, 2020; Lamers et al, 2020). Gastrointestinal symptoms have been linked to prolonged infection and a more severe course of disease (Zhong et al, 2020). The entry receptor for SARS-CoV-2 is angiotensin-converting enzyme 2 (ACE2) expressed on epithelial cells. Spike glycoproteins present in the viral envelope bind to ACE2 and are proteolytically cleaved by the transmembrane protease serine protease 2 (TMPRSS2), which results in virus entry (Hoffmann et al, 2020; Matsuyama et al, 2020; Yan et al, 2020).

Virus infection of epithelial cells triggers an inflammatory response. While a well-coordinated immune response leads to viral clearance, excessive, dysfunctional responses are incapable of clearing the infection and are linked to severe COVID-19 symptoms (Zheng et al, 2020). It is not clear why such immune responses are inefficient and whether and how SARS-CoV-2 undermines the immune response. IFN-γ is a central antimicrobial cytokine that has been shown to be upregulated in the context of COVID-19, both locally in the mucosa and systemically (Chua et al, 2020; Huang et al, 2020). It binds to IFN-γ receptors expressed on various cell types (Hu & Ivashkiv, 2009) and orchestrates the cellular immune responses to infection by multiple means such as activation of macrophages, enhanced antigen presentation, and T cell differentiation (Borden et al, 2007). In addition, IFN-γ can also signal to epithelial cells directly, leading to different responses such as expression of chemokines and secretion of antimicrobial proteins (Farin et al, 2014; Walrath et al, 2020). Recent reports have revealed an increase in IFN-γ signaling upon SARS-CoV-2 infection in lung epithelial cells (Chua et al, 2020). Moreover, using scRNAseq from patient airway samples, an association between

1 Department of Hepatology and Gastroenterology, Charité University Medicine, Berlin, Germany
2 Department of Molecular Biology, Max Planck Institute for Infection Biology, Berlin, Germany
3 Berlin Institute for Medical Systems Biology, Max Delbrück Center for Molecular Medicine, Berlin, Germany
4 Institute of Virology, Freie Universität Berlin, Berlin, Germany
5 Department of Surgery, Charité University Medicine, Berlin, Germany
6 Department of Infectious Disease and Public Health, Jockey Club College of Veterinary Medicine and Life Sciences, City University of Hong Kong, Kowloon, Hong Kong
*Corresponding author. Tel: +49 (0)30 450 553022; E-mail: michael.sigal@charite.de

interferon signaling and ACE2 expression levels has been suggested (Ziegler *et al*, 2020). However, mechanisms by which IFN-γ regulates ACE2 expression and functional data on how it affects SARS-CoV-2 are lacking.

Here, we applied a primary 3D organoid system from the colon that mimics epithelial turnover and differentiation. We demonstrate that IFN-γ is a strong and efficient driver of epithelial differentiation into enterocytes that express high levels of ACE2. IFN-γ-induced, differentiated colonocytes are highly susceptible to SARS-CoV-2 infection, and we demonstrate virus replication and release from dying colonocytes. Moreover, we demonstrate that SARS-CoV-2 itself triggers epithelial IFN responses that are sufficient to induce increased differentiation and ACE2 expression. Our data reveal how SARS-CoV-2 exploits a central immune response pathway for epithelial invasion and replication, which may have a significant impact on the clinical course of disease as well as on viral transmission.

# Results

### IFN-γ is a strong driver of ACE2 expression in human colon organoids

ACE2 is the primary entry receptor for SARS-CoV-2 and a large proportion of COVID-19 patients experience gastrointestinal symptoms. We thus explored the expression of ACE2 in human colon tissue by analyzing single-cell RNA sequencing data of a recently published data set (Parikh *et al*, 2019). The results demonstrate that surface colonocytes expressing *KRT20* and *AQP8*, both markers of differentiated enterocytes (Fig 1A), also express ACE2 as well as the receptor for IFN-γ (IFNGR2). Immunofluorescence of human colon tissue indeed confirmed ACE2 expression at the apical side of the surface epithelial cells (Fig 1B).

SARS-CoV-2-infected individuals show elevated levels of IFN-γ that may influence the progression of COVID-19 (Chua *et al*, 2020; Huang *et al*, 2020). To explore the potential connection between IFN-γ and ACE2, we established cultures of human colon organoids (hCOs). hCOs grown in full medium (supplemented with the growth factors WNT, Noggin and EGF) showed signs of self-organization and development of crypt-like buds (Fig 1C, left panel). When organoids were treated with IFN-γ for 3 days, we observed an increased abundance of KRT20-positive cells (Fig 1C). Switching from full medium to basic medium (by removal of the supplemented growth factors) for 4 days directed cells to a more differentiated state, which was evident by the pronounced increase in KRT20$^+$ cells as well as an increase in signal intensity, as assessed by confocal microscopy (Figs 1C and D and EV1A). hCOs grown in basic medium showed a robust IFN-γ response between 5 and 100 ng/ml, as assessed by qRT–PCR for expression of the IFN-γ target gene *IFIT3* (Fig EV1B). We used the intermediate concentration of 50 ng/ml for subsequent experiments and utilized E-cadherin staining to better visualize the cell shape and performed actin staining to show that upon IFN-γ treatment cells become more columnar, a feature of differentiation (Fig EV1C). Of note, IFN-γ treatment reduced the proportion of goblet cells in the presence of both full and basic medium conditions, as assessed by alcian blue staining (Fig EV1D). To quantify the effects of IFN-γ on differentiation, we analyzed mRNA expression of the enterocyte marker *KRT20* and *ALPI*, the

goblet cell markers *MUC2 and ITF*, the enteroendocrine cell marker *CHGA*, and the stem cell marker *LGR5*. In basic medium conditions, the differentiation markers for absorptive and secretory cells increased (Figs 1E–G and EV1E), while the stem cell marker *LGR5* was strongly decreased (Fig 1H). IFN-γ treatment increased *KRT20* as well as *ALPI* expression, while the secretory cell markers were reduced only in basic medium (Figs 1E–G and EV1E) confirming the histological findings. Concomitantly, expression of *ACE2* was also significantly increased upon IFN-γ treatment in differentiated organoids (Fig 1I). Immunofluorescence imaging revealed expression of ACE2 at the apical side of differentiated human organoids, which was further increased by additional treatment with IFN-γ (Fig 1J). In mouse colon, Ace2 is also expressed in Krt20$^+$ surface colonocytes (Fig EV2A). Single-cell RNA sequencing data from a recently published data set accessible by an online tool (Tabula Muris Consortium *et al*, 2018) showed that the receptors for IFN-γ (*INFGR1/2*) are expressed in *Krt20* expressing enterocytes (Fig EV2B). IFN-γ treatment of mouse colon organoids induced differentiation, similar to that induced by removal of the Wnt signaling factors sWnt and CHIR from the medium (Fig EV2C). IFN-γ treatment resulted in more columnar shaped cells (Fig EV2C) that express increased levels of Krt20 protein (Fig EV2D) and show reduced proliferation (Fig EV2E). qPCR analyses from organoids treated with IFN-γ and corresponding controls confirmed a significant increase in *Krt20* as well as *Ace2* expression (Fig EV2F). We conclude that IFN-γ rapidly induces a differentiation program in organoids toward the enterocyte lineage that is conserved between species.

### IFN-γ increases susceptibility of colon epithelial cells to SARS-CoV-2 infection

It has recently been demonstrated that human small intestinal organoids can be productively infected by SARS-CoV-2 (Lamers *et al*, 2020). To explore whether the increased *ACE2* expression affects the efficiency of infection with SARS-CoV-2, we cultured human colon organoids in full as well as basic medium, with or without IFN-γ treatment and infected them with SARS-CoV-2 at an MOI of 1. Immunofluorescence staining for the nucleocapsid protein (N-protein), indicative of productive infection, showed infected cells in all conditions (Fig 2A). In organoids grown in full medium, however, only some cells were positive for N-protein, whereas almost all cells were infected in cells grown in basic medium. Moreover, IFN-γ appeared to increase the proportion of infected cells as well as N-protein fluorescence intensity in both conditions (Fig 2A). To quantify virus replication, we measured virus RNA levels by qPCR. Organoids cultured in basic medium had a higher virus load compared with cells grown in full medium (Fig 2B), and IFN-γ-treated cells had a higher virus load in both medium conditions (Fig 2C and D), which was most pronounced in organoids cultured in basic conditions (Fig 2E). Comparing the virus load at 24 and 48 h post-infection revealed an increase in SARS-CoV-2 over time in IFN-γ treated cells, demonstrating efficient viral replication (Fig 2F). We also grew colon cells as polarized, 2-dimensional air–liquid interface (ALI) cultures and infected them from the apical side with SARS-CoV-2, confirming susceptibility of colon cells in this system (Fig EV3A). However, ALI cultures showed less robust growth and already contained high levels of differentiated cells in full medium

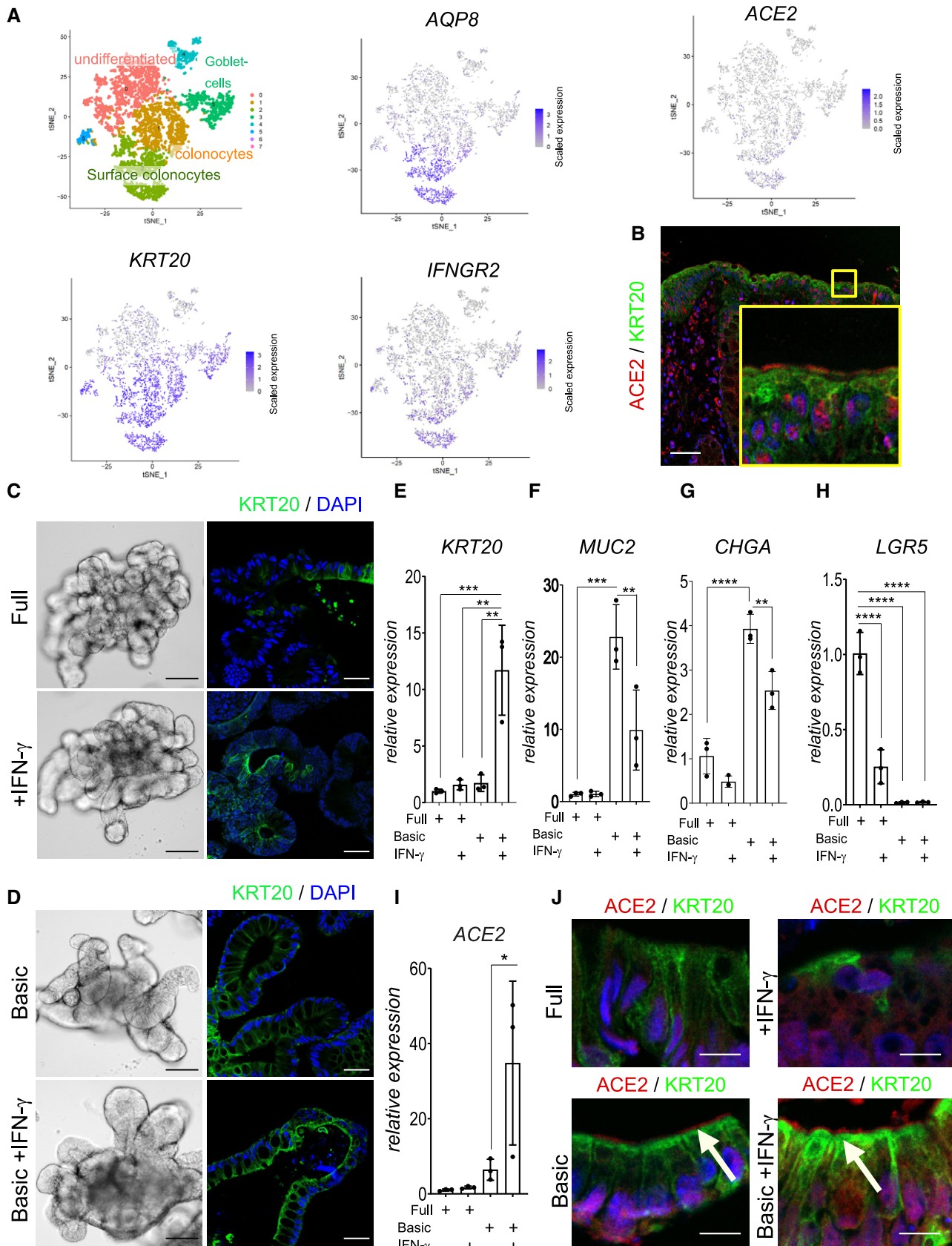

**Figure 1.**

**Figure 1.  IFN-γ is a strong driver of ACE2 expression in human colon organoids.**

A   (left) t-sne plot from scRNAseq data of human organoids and cluster assignment of colonic cell populations. Red color corresponds to undifferentiated cells and light green corresponds to surface colonocytes (enterocytes). t-sne plot for selected genes from single-cell RNAseq of human colon epithelial cells revealing expression of ACE2 specifically in differentiated surface colonocytes expressing KRT20 and AQP8, which also express the IFN-γ receptor IFNGR2.

B   Immunofluorescence staining for ACE2 (red) and KRT20 (green) of human colon tissue section indicating ACE2 expression in surface colonocytes. Scale bar: 50 μm.

C   Human colon organoids cultured in full medium (FM) condition (upper panel) or FM treated with IFN-γ (lower panel). Left: bright field images of 7-day cultured human colon organoids, scale bar: 100 μm. Right: Immunofluorescence for KRT20 (green; note increased expression upon IFN-γ treatment). Scale bar: 25 μm.

D   Human colon organoids cultured in basic medium (upper panel) or additionally treated with IFN-γ (lower panel). Left: bright field images of organoids that were cultured for 4 days in basic medium and in addition for 3 days with IFN-γ (lower panel) scale bar: 100 μm. Right: Immunofluorescence labeling for KRT20 (green). Scale bar: 25 μm.

E–I   Comparison of E) KRT20, F) MUC2, G) CHGA, H) LGR5, and I) ACE2 mRNA expression in organoids grown in FM or in basic medium and either untreated or treated with IFN-γ (n = 3). Data are presented as mean ± SD, *: $P \leq 0.05$, **: $P \leq 0.001$, ***: $P \leq 0.0001$, as determined by one-way ANOVA, followed by Tukey's multiple comparisons test (more details see Appendix Table S2).

J   Immunofluorescence staining for ACE2 (red) and KRT20 (green) of organoids grown in full medium (upper) or basic medium (lower) and additionally treated with IFN-γ (right), indicating expression of ACE2 in differentiated enterocytes. White arrows point to ACE2. Scale bar: 10 μm.

conditions, overall indicating that 3D organoids are more appropriate for studying effects of differentiation and infection with SARS-CoV-2. We performed transmission electron microscopy on the IFN-γ-treated organoids 48 h after infection and observed a high number of particles of ~ 100 nm in diameter, phenotypically resembling the morphology of a coronavirus. The particles were found in large intracellular vesicles, which were randomly distributed but in proximity to other cellular components, such as mitochondria and rough endoplasmic reticulum (Fig 2G). Further, we observed cells heavily loaded with virus particles (Fig 2G lower and inset). Such cells showed signs of cell lysis and some were removed from the epithelial structure of the organoids, consistent with the light microscopy images of organoids at 48 h after infection, which showed signs of disintegration (Fig EV3B). Immunofluorescence for cleaved caspase-3, a marker of apoptosis, confirmed increased cell death in infected, differentiated organoids (Fig 2H, quantification below). Thus, our data indicate that SARS-CoV-2 can invade and efficiently replicate in IFN-γ-treated enterocytes. Upon infection, these cells produce large numbers of virus particles, ultimately leading to disintegration and apoptosis.

To explore how infection affects the epithelium, we performed a transcriptome analysis of organoids grown in basic medium 48 h after infection and compared the results with uninfected controls. The comparative analysis revealed a strong upregulation of several IFN-γ target genes (Table EV1). Gene set enrichment analysis confirmed the most significant positive enrichment for GO gene sets linked to virus infection such as "GO response to virus" as well as gene sets linked to IFN signaling such as "GO responses to type I interferon" as well as "GO response to interferon gamma" (Fig 3A). Similarly, we observed a highly significant positive enrichment for the "Hallmark Interferon gamma response" gene set (Fig 3B). To compare the cellular responses of colon organoids to infection with those induced by IFN-γ, we also analyzed the transcriptome of organoids treated with IFN-γ compared with uninfected controls (Table EV2). Not unexpectedly, we found various IFN-γ target genes to be upregulated. Comparison of differentially expressed genes of SARS-CoV-2-infected and IFN-γ-treated organoids using DISCO revealed that several of the top regulated genes were induced by both treatments (Fig 3C). Furthermore, when we analyzed the 20 genes with the highest fold-change increase upon infection with SARS-CoV-2, we noticed that 18 of them were also significantly induced by IFN-γ treatment (Appendix Table S1). Overall, this

indicates that infection triggers signaling pathways in the epithelium that, to a large extent, resemble epithelial responses to IFN-γ. Accordingly, analysis of enterocyte-specific genes following SARS-CoV-2 infection revealed that enterocyte marker genes, including ACE2 and KRT20 were also significantly increased (Fig 3D). Under conditions of differentiation, all cells already expressed KRT20; however, SARS-CoV-2 infection further increased KRT20 expression, as observed by immunofluorescence (Fig 3E). We used qPCR to validate the SARS-CoV-2 infection-dependent upregulation of the IFN-γ target genes CXCL11 and IFIT3, as well as KRT20 and ACE2 (Fig 3F) and indeed confirmed an upregulation of these genes upon infection. IFN-γ induces JAK/STAT signaling, which can be diminished by the pan-JAK inhibitor pyridone 6 (P6) (Nakagawa et al, 2011). Indeed organoids infected with SARS-CoV-2 show reduced expression of IFIT3 and CXCL11 as well as ACE2, KRT20 when treated with P6 (Fig EV4A). Lastly, we asked whether JAK/STAT activity may have an impact on infectivity of organoids and to this end infected organoids treated either with IFN-γ alone or with IFN-γ and pyridine. IFN-γ treatment increased the virus load of organoids infected with SARS-CoV-2, while the JAK inhibition by P6 prevented this increase (Fig EV4B).

## Discussion

Here, we demonstrate that differentiated enterocytes from human colon express ACE2 and are susceptible to SARS-CoV-2 infection. We also reveal that IFN-γ is a strong driver of epithelial differentiation toward the enterocyte lineage, resulting in high ACE2 expression and increased susceptibility to SARS-CoV-2. Moreover, the transcriptional response of epithelial cells to infection indicates a strong activation of IFN signaling, resembling IFN-γ treatment, and also promoting differentiation. These data suggest that infection-driven inflammation may create a vulnerable state, which in turn enables robust virus replication and release, which may have consequences for the clinical course of the disease as well as for virus transmission.

We used organoids derived from the large intestine (colon) and confirm recent reports that used small intestine to demonstrate productive infection with SARS-CoV-2 (Lamers et al, 2020; Zhou et al, 2020). Gastrointestinal infection by SARS-CoV-2 and its role in COVID-19 is an emerging area of investigation (Gupta et al, 2020).

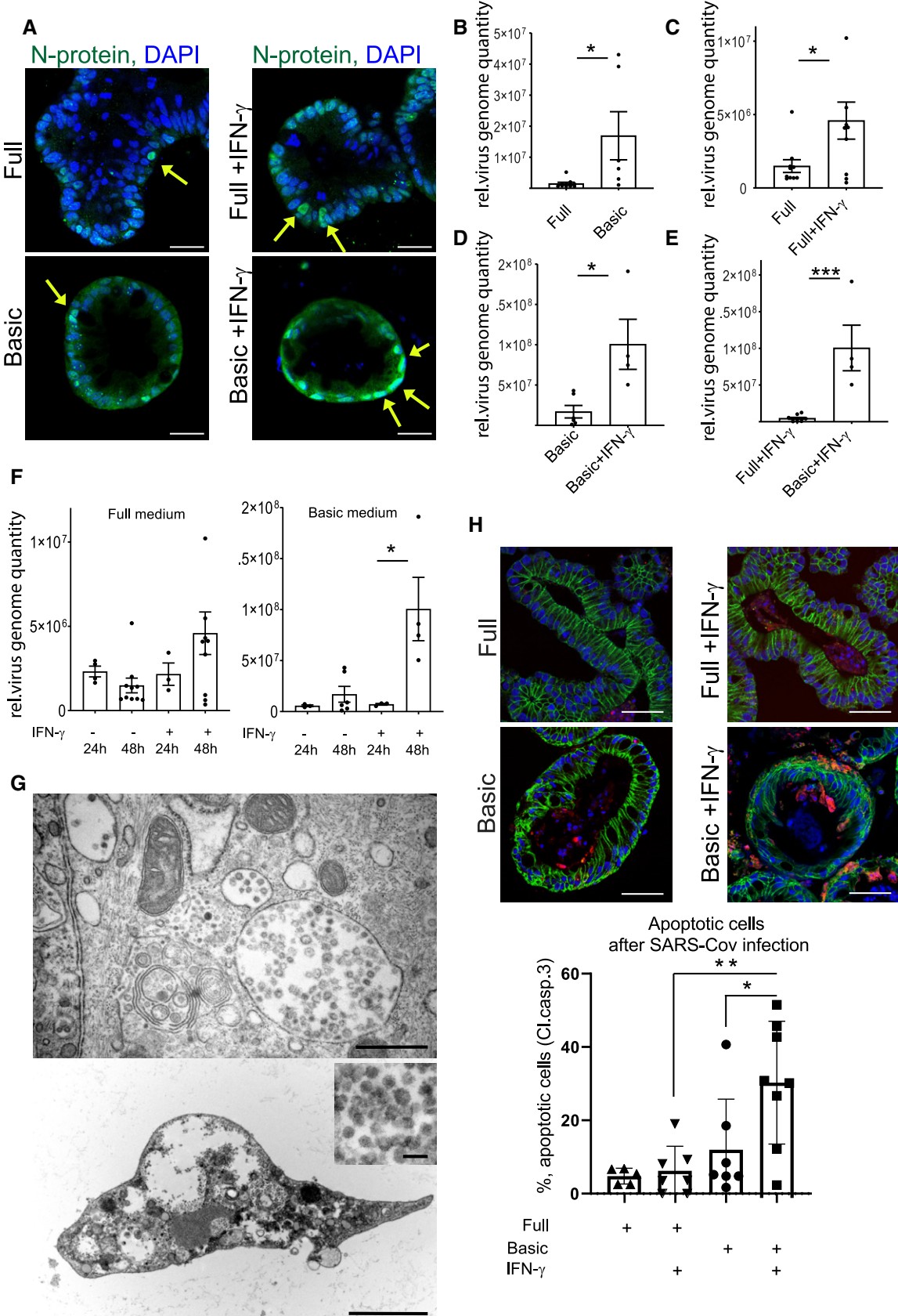

**Figure 2.**

**Figure 2.   IFN-γ increases infectivity of SARS-CoV-2 in colonic organoids.**

A   Immunofluorescence for N-protein (green) 48 h after SARS-CoV-2 infection of human colon organoids cultured in full medium (upper), full medium + IFN-γ (upper right), basic medium (lower left), and basic medium + IFN-γ (lower right). Yellow arrows point to N-protein-positive cells. Scale bar: 25 μm.

B   qPCR data displaying the relative virus load of SARS-CoV-2 measured by viral genome quantity in full medium vs. basic medium organoids 48 h after infection and normalized to *GAPDH* (n = 6). Data are presented as mean ± SD, *: $P \leq 0.05$, **: $P \leq 0.001$, ***: $P \leq 0.0001$, as determined by Student's *t*-test (more details see Appendix Table S2).

C   qPCR data displaying the relative virus load of SARS-CoV-2 measured by viral genome quantity in full medium vs. full medium + IFN-γ-treated organoids 48 h after infection and normalized to *GAPDH* (n = 9). Data are presented as mean ± SD, *: $P \leq 0.05$, **: $P \leq 0.001$, ***: $P \leq 0.0001$, as determined by Student's *t*-test (more details see Appendix Table S2).

D   qPCR data displaying the relative virus load of SARS-CoV-2 measured by viral genome quantity in basic medium vs. basic medium + IFN-γ-treated organoids 48 h after infection and normalized to *GAPDH* (n = 4). Data are presented as mean ± SD, *: $P \leq 0.05$, **: $P \leq 0.001$, ***: $P \leq 0.0001$, as determined by Student's *t*-test (more details see Appendix Table S2).

E   qPCR data displaying the relative virus load of SARS-CoV-2 measured by viral genome quantity in full medium + IFN-γ vs. basic medium + IFN-γ-treated organoids 48 h after infection and normalized to *GAPDH* (n = 4). Data are presented as mean ± SD, *: $P \leq 0.05$, **: $P \leq 0.001$, ***: $P \leq 0.0001$, as determined by Student's *t*-test (more details see Appendix Table S2).

F   qPCR data comparing the relative virus load of SARS-CoV-2 measured by viral genome quantity normalized to *GAPDH* at 24 and 48 h after infection in full medium (left) and basic medium (right) untreated or treated with IFN-γ, indicating increase in virus load in differentiated and IFN-γ-treated conditions (n = 8). Data are presented as mean ± SD, *: $P \leq 0.05$, **: $P \leq 0.001$, ***: $P \leq 0.0001$, as determined by one-way ANOVA, followed by Tukey's multiple comparisons test (more details see Appendix Table S2).

G   Electron microscopy imaging of organoids grown in basic medium, pretreated with IFN-γ for 3 days and infected for 48 h with SARS-CoV-2. Upper: Overview showing large virus-loaded vesicles in an epithelial cell. Scale bar: 500 nm. Lower: Disintegrated organoid cell containing high virus load, scale bar: 2.5 μm, higher magnification to visualize virus particles, scale bar: 100 nm.

H   Immunofluorescence for cleaved caspase-3 (red) and E-cadherin (green) 48 h after SARS-CoV-2 infection of human colon organoids cultured in full medium (upper), full medium + IFN-γ (upper right), basic medium (lower left) and basic medium + IFN-γ (lower right). Scale bar: 50 μm. (below) Quantification of apoptotic cells, indicating increased cell death in the differentiated and IFN-γ-treated condition after SARS-CoV-2 infection (n = 7). Data are presented as mean ± SD, *: $P \leq 0.05$, **: $P \leq 0.001$, ***: $P \leq 0.0001$, as determined by one-way ANOVA, followed by Tukey's multiple comparison test (more details see Appendix Table S2).

Gastrointestinal symptoms have been observed in a large proportion of patients with SARS-CoV-2. Immunofluorescence staining of samples from patients with COVID-19 suggests that their glandular gut epithelium can be infected with SARS-CoV-2 (Xiao *et al*, 2020), implicating a potential fecal-oral transmission route, which has been further substantiated by the notion that viral RNA can be detected in feces of around 50% of patients with COVID 19, even after respiratory recovery (Cheung *et al*, 2020). Thus, understanding the interaction of gastrointestinal epithelia with SARS-CoV-2 is important for uncovering the mechanism of SARS-CoV-2 infection and COVID-19 disease. In this context, organoids represent a valuable tool to not only explore cellular interactions with SARS-CoV-2 but also mimic in vivo aspects such as cellular heterogeneity within epithelial units, niche signals that regulate epithelial turnover and differentiation, and the impact of immune responses on epithelial behavior and viral replication.

IFN-γ is a central mediator of host responses to pathogens and a plethora of IFN-γ driven antiviral mechanisms have been demonstrated. In the context of COVID-19, IFN-γ signaling has been found to be upregulated in the airway mucosa of infected individuals (Chua *et al*, 2020). IFN-γ is secreted by different subsets of immune cells, particularly Th1 T cells, and is a master orchestrator of the immune response to infection. Our data demonstrate that IFN-γ also has a direct effect on epithelial cells. In addition to activating epithelial immune response mechanisms, including increased expression of genes involved in antigen presentation and chemokine production, we show that IFN-γ alters epithelial cell fate determination. A direct effect of T helper cell cytokines on epithelial stem cell fate determination and, therefore, cellular composition in the gastrointestinal tract has recently been demonstrated (Biton *et al*, 2018), establishing that epithelial differentiation is shaped by the immune environment. Similarly, our data suggest that IFN-γ drives differentiation toward the enterocyte lineage. It was previously reported that extended exposure to IFN-γ can interfere with Wnt signaling by

inducing expression of Dkk1 (Nava *et al*, 2010). Such interference of IFN-γ with Wnt signaling activity may explain the increased cell fate switch toward the enterocyte linage that we observed in mouse and human colon organoids. In the colon, ACE2 is specifically expressed in surface enterocytes and the upregulation upon IFN-γ treatment we observed is at least partially due to altered differentiation of organoids. In the lung, ACE2 expression also seems to be regulated by IFN-γ, which may be similarly due to IFN-driven cellular cell fate determination (Ziegler *et al*, 2020). It will be important to explore how IFN-γ affects lung cell differentiation and whether its differentiation-promoting effects are responsible for enhanced ACE2 expression in the airway system.

Our data showing how IFN-γ signaling can promote virus replication in mucosal tissue may explain several clinical observations related to COVID-19. Indeed, severe cases of COVID-19 are linked to strong IFN responses that appear to be inefficient with respect to clearance of the infection. A recent report that evaluated single-cell RNA expression data from the airways of patients with COVID-19 revealed a correlation between the severity of SARS-CoV-2 infection and ACE2 expression in response to IFN-γ produced by immune cells (Chua *et al*, 2020). While IFN-γ is likely important in the context of the host immune response to SARS-CoV-2 infection, its effects on epithelial differentiation and ACE2 expression may explain why it is inefficient in limiting infection in some patients. Several clinical trials are currently investigating the outcome of SARS-CoV-2 infection as a function of activation of IFN signaling as well as suppression of this pathway via Jak-Stat inhibitors (Satarker *et al*, 2020). These studies will reveal whether, in a clinical setting, this signaling pathway induces immunity and is beneficial, or whether the virus highjacks IFN signaling through the mechanisms described here. In case of the latter, inhibition of this pathway may reduce virus replication and severity of COVID-19. In addition to having a potential impact on the clinical outcome, IFN-γ may also have an impact on virus transmission, as our data suggest that

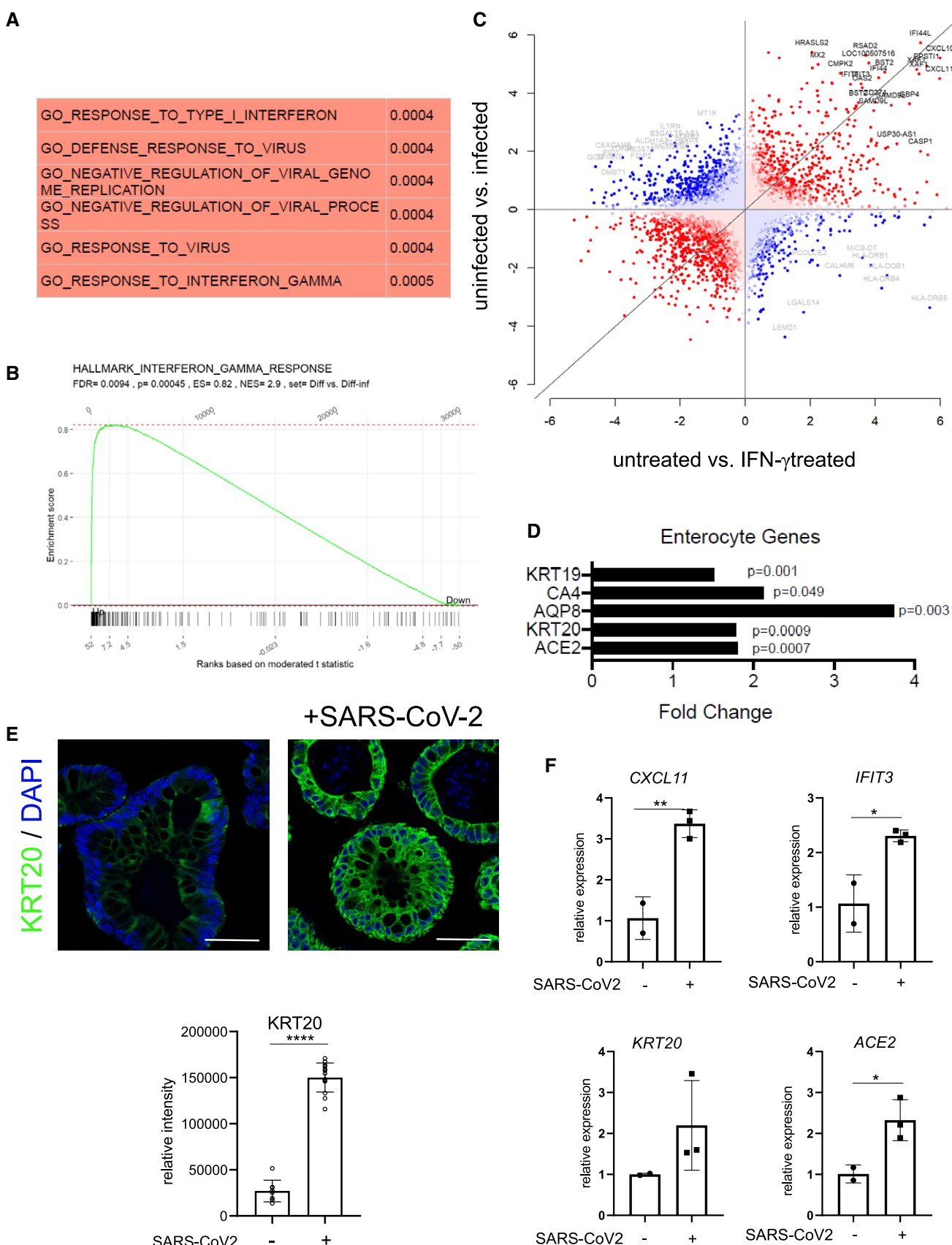

Figure 3.

**Figure 3.   SARS-CoV-2 induces IFN-γ gene expression signature and promotes differentiation.**

A   Gene set enrichment analysis of the transcriptome data from uninfected vs. 48 h SARS-CoV-2 infected organoids was performed and the GO terms with the most significant positive correlation are displayed.

B   Gene set enrichment analysis of the transcriptome data comparing uninfected vs. 48 h SARS-CoV-2-infected organoids and the "Hallmark interferon gamma response" gene set. *P*-values were calculated using the fgsea package from https://www.bioconductor.org with 5,000 permutations.

C   Analysis of discordance and concordance of transcriptomic responses (DISCO) between untreated vs. SARS-CoV-2-infected and untreated vs IFN-γ-treated organoids. The 50 most regulated genes are highlighted; gray: discordant and black: concordant.

D   Relative expression of selected enterocyte-specific genes upon SARS-CoV-2 infection, normalized to *GAPDH*. Differences in gene expression were assessed using the linear model 'lmFit' and 'makeContrasts' in limma.

E   Immunofluorescence for KRT20 (green) in human colon organoids cultured in basic medium and either non-infected (left) or 48 h after SARS-CoV-2 infection (right), indicating a strong increase in KRT20 expression after viral infection. Scale bar: 50 μm. Quantification of fluorescence intensity below (*n* = 8). Data are presented as mean ± SD, ****: *P* ≤ 0.0001, as determined by Student's *t*-test (more details see Appendix Table S2).

F   qPCR data displaying the increase of INF-γ target genes (*CXCL11*, *IFIT3*) (upper) and enterocyte marker genes (*KRT20*, *ACE2*) (lower) after SARS-CoV-2 infection, validating the results from the microarray analysis (*n* = 3). Data are presented as mean ± SD, ****: *P* ≤ 0.0001, as determined by Student's *t*-test (more details see Appendix Table S2).

differentiated cells have high virus loads that are released upon cellular disintegration.

Differing IFN-γ responses to SARS-CoV-2 may also be responsible for the highly variable clinical outcome of COVID-19. For example, IFN-γ levels appear to be age-related, with the elderly showing higher IFN-γ production compared with younger subjects (Bandres *et al*, 2000; Yen *et al*, 2000), correlating with the increased severity of COVID-19 observed in elderly patients (Jordan *et al*, 2020). Similarly, it was recently reported that the chronic inflammatory state in the lungs of smokers increases the risk for severe course and adverse outcomes of COVID-19 (Vardavas & Nikitara, 2020; Zhou *et al*, 2020). Our data suggest that the virus' ability to increase its replication efficiency via IFN-γ signaling could contribute to the increased severity of SARS-CoV-2 infection and disease progression of COVID-19 in patients with chronic inflammation.

In summary, our data reveal a new and somewhat counterintuitive mechanism by which SARS-CoV-2 promotes infection of epithelial cells in the context of inflammation, which may explain the limited efficiency of the immune system to control viral replication. This function of IFN-γ and downstream signaling may contribute to the unusual pathogenesis of COVID-19 and transmission of SARS-CoV-2.

# Materials and Methods

### Primary human colon organoid culture

Colon biopsies were obtained from Charité University Hospital, Berlin, under signed informed consent obtained from all human subjects. All experiments conformed with the WMA Declaration of Helsinki and the Department of Health and Human Services Belmont Report. The study was approved by the ethics committee of the Charité University Medicine, Berlin (EA2/008/18). Fresh colon samples from surgical resections of the sigmoid colon were washed five times in ice cold phosphate-buffered saline (PBS) supplemented with 1× penicillin/streptomycin (#15140122, Gibco), 50 μg/ml Gentamicin (#G1272, Merck), and 2.5 μg/ml amphotericin B (#Y0000005, Sigma) and incubated for 15 min at room temperature. The mucosa was cut into approx. 0.5 cm² pieces and incubated for 5 min in 10 mM EDTA/PBS. Pieces of tissue were transferred to 2 mM EDTA/PBS supplemented with 2.5 μM DTT (#D9779, Sigma) and incubated on a rotating shaker for 40 min at 4°C. After transfer

to PBS and vigorous shaking, the crypt-containing supernatant was transferred to 0.1% BSA/PBS and centrifuged at 200 *g* for 3 min at 4°C. The number of colonic crypts was determined and 10 crypts per 1 μl Matrigel drop (#356231, Corning) seeded in a volume of 15 μl in 48-well plates.

Organoids were grown based on previously developed protocols (Sato *et al*, 2011; Michels *et al*, 2019) with further modifications. Human colon organoid lines were cultured in basic medium composed of Advanced Dulbecco's Modified Eagle's Medium/F12 (#12634, Gibco) containing 10 mM HEPES (#15630056, Gibco), 2 mM GlutaMAX (#35050061, Gibco), 1.25 mM N-acetylcysteine (#A9165, Sigma), 25% R-spondin1 conditioned medium, 1× B-27 (#17504044, Gibco), 1× N2 (#17502048, Gibco), 500 nM A83-01 (#616454, Merck), 100 μg/ml primocin (#ant-pm, Invivogen), and 10% penicillin/streptomycin (#15140122, Gibco), which was supplemented with 1 mM nicotinamide (N0636, Sigma), 10 μM SB202190 (#S7067, Sigma), 3 μM CHIR-99021 (#S1263, Selleckchem), 0.328 nM sWnt (#N001, U-Protein Express), 50 ng/ml hEGF (#PHG0311, Invitrogen), and 100 ng/ml hNoggin (#120-10C, PeproTech) for maintenance in complete medium conditions. 10 μM Y-27632 (#M1817, Hölzel) was added after passaging and after the initial seeding. After 4 days of culture in complete medium, differentiation was induced by further culturing in basic medium. Human organoids were passaged as organoid-crypt-fragments generated by passing 10 times through a 26-G needle. IFN-γ treatment was carried out with 40 ng/ml human IFN-γ (#285-IF-100, R&D) for the indicated times. To inhibit Jak/Stat signaling, organoids were treated with 10 μM pyridone-6 (P6).

### Primary mouse colon organoid culture

Mouse colon was dissected, opened longitudinally, and cut into 5 mm long pieces. Tissue was washed five times in ice cold PBS, followed by incubation for 5 min in 10 mM EDTA/PBS. Pieces of tissue were transferred to 2 mM EDTA/PBS supplemented with 2.5 μM DTT and incubated on a rotating shaker for 20 min at 4°C. Buffer was changed to HBSS (#14025050, Gibco) and transferred to S-Tubes and run in a gentleMACS Dissociator (Miltenyi) using the Spleen2.01 program seven times. The crypt-containing supernatant was transferred to 0.1% BSA/PBS and centrifuged at 200 *g* for 3 min at 4°C. The number of colonic crypts was determined and 20 crypts per 1 μl Matrigel (#356231, Corning) seeded in a volume of 15 μl in 48-well plates.

Mouse colon organoid lines were cultured in basic medium consisting of Advanced Dulbecco's Modified Eagle's Medium/F12 (#12634, Gibco) containing 10 mM HEPES (#15630056, Gibco), 2 mM GlutaMAX (#35050061, Gibco), 1.25 mM N-acetylcysteine (#A9165, Sigma), 25% R-spondin 1 conditioned medium, 1× B-27 (#17504044, Gibco), 1× N2 (#17502048, Gibco), 500 nM A83-01 (#616454, Merck), and 10% penicillin/streptomycin (#15140122, Gibco), which was supplemented with 3 μM CHIR-99021 (#S1263, Selleckchem), 0.328 nM sWnt (#N001, U-Protein Express), 50 ng/ml mEGF (#PMG8043 Invitrogen), and 100 ng/ml mNoggin (#250-38, PeproTech) for maintenance in complete medium conditions. 10 μM Y-27632 (#M1817, Hölzel) was added after passaging and after the initial seeding. After 3 days of culture in full medium, differentiation was induced by further culturing either in basic medium, or supplemented with sWnt and CHIR or with mEGF and mNoggin. Mouse colon organoids were passaged as single cells, which were generated by 6-min incubation in TriplE (#12604013, Gibco) at 37°C. IFN-γ treatment was carried out with 100 ng/ml mouse IFN-γ (#485-MI, R&D) for the indicated times.

## Immunofluorescence staining and imaging

Organoids were transferred to 3.7% formaldehyde and fixed for 3 h at room temperature, followed by incubation in 0.1% BSA/PBS for at least 30 min, embedding in 2% agarose, dehydration, and embedding in paraffin. 5 μm sections were deparaffinized and rehydrated, followed by antigen retrieval in citrate buffer. Non-specific antibody binding was blocked by incubation in 0.1% tween/PBS supplemented with 5% FBS and 1% BSA for 30 min, followed by overnight incubation with primary antibodies detecting KRT20 (#13063, Cell Signaling), anti-E-cadherin (#610181, BD), N-protein (Bussmann et al, 2006), ACE2 (#HPA000288, Sigma), or Ki67 (#9192, Cell Signaling). The next day, they were incubated with secondary antibodies (Alexa488, Cy3 or Cy5) (Jackson Immunoresearch) for 2 h at room temperature and counterstained with DAPI. Immunofluorescence imaging procedures were acquired with a laser scanning microscope LS8 (Leica, Germany), and images were analyzed with Fiji (ImageJ).

## RNA isolation and RT–PCR

Organoid medium was removed and RNA was isolated using the Macherey & Nagel RNA isolation kit (#740955) according to the manufacturer's instruction. cDNA was generated using the iScript cDNA synthesis kit (#1708891, Bio-Rad), and qPCR was performed with SYBR-green (#A25741, Thermo Fisher) and the StepOne Real-Time PCR System (Thermo Fisher). Fold-change expression was determined following the deltaCC method and normalized to GAPDH. Viral RNA was detected by TaqMan RT–PCR as described before (Corman et al 2020) and normalized to GAPDH (ID: sh99999905m1).

## Virus infection

A human SARS-CoV-2 isolate (BetaCoV/Germany/BavPat1/2020 (Wolfel et al, 2020) was obtained as a generous gift from Dr. Daniela Niemeyer and Prof. Christian Drosten, Charité—University Medicine Berlin. The virus was propagated on Vero E6 cells (ATCC CRL-1586) in minimal essential medium (MEM; #P04-09500, PAN

Biotech, Aidenbach, Germany) supplemented with 10% fetal bovine serum (#P30-3031, PAN Biotech), 100 IU/ml penicillin G (HP49.1), and 100 g/ml streptomycin (#HP66) (Carl Roth, Karlsruhe, Germany). All work involving SARS-CoV-2 was performed under the appropriate safety precautions in a BSL-3 facility (Freie Universität Berlin, Institute of Virology). Virus containing cell culture supernatant was harvested when cytopathic effects (cpe) became apparent (typically 48 h following infection) and stored at −80°C. Titrations were performed on Vero E6 cells under a 1.3 % methylcellulose (#M0262, viscosity 400 cP, Sigma, St. Louis, MO, USA) overlay. Plaque-forming units were counted following fixation with 4% formalin and crystal violet staining at 72 h post-infection.

Organoids were cultured in complete medium for 3 days and further cultured either in complete medium or differentiation medium for 4 days, followed by 3 days of IFN-γ treatment. Organoids were harvested and broken into cell clusters by passaging through a 26-G needle and washed in 0.1% BSA/PBS buffer. A sample was incubated for 6 min in TryplE (#12604013, Gibco) at 37°C to generate single-cell suspension to determine cell numbers. For virus infection, cell clusters were incubated with supernatant from SARS-CoV-2-infected Vero E6 cell cultures at virus titer of $4 \times 10^6$ resulting in a multiplicity of infection (MOI) of 1. Excess virus was removed by three washes in 0.1% BSA/PBS buffer before seeding in Matrigel. Organoids were cultured for up to 48 h post-infection in the respective medium.

## Microarray

Microarray experiments were performed as independent dual-color dye-reversal color-swap hybridizations. Total RNA was labeled with the Low Input Quick-Amp Labeling Kit (Agilent Technologies). In brief, 100 ng total RNA was reverse transcribed and amplified using an oligo-dT-T7 promoter primer and labeled with cyanine 3-CTP and/or cyanine 5-CTP by T7 in vitro transcription. After precipitation, purification, and quantification, 0.5 μg labeled cRNA of each ratio sample was mixed, fragmented, and hybridized to custom whole genome human 8 × 60K multipack microarrays (Agilent-048908) according to the supplier's protocol (Agilent Technologies). Scanning of microarrays was performed with 3 μm resolution and 20-bit image depth using a G2565CA high-resolution laser microarray scanner (Agilent Technologies). Microarray image data were processed with the Image Analysis/Feature Extraction software G2567AA v. A.12.1.1.1 (Agilent Technologies) using default settings and the GE2_1200_Dec17 extraction protocol.

The extracted MAGE-ML files were analyzed with the Rosetta Resolver Biosoftware, Build 7.2.2 SP1.31 (Rosetta Biosoftware). Ratio profiles comprising single hybridizations were combined in an error-weighted fashion to create ratio experiments. A 0.5 log2 fold-change expression cut-off for ratio experiments was applied together with anti-correlation of ratio profiles, rendering the microarray analysis highly significant ($P < 0.05$).

In addition, microarray data were analyzed using the R package limma (Ritchie et al, 2015). The microarray readout txt files were background corrected, normalized, and controlled for quality using the R package limma (Smyth & Speed, 2003). Between-array normalization was done using the quantile method in limma. The hybridization control samples were removed and the gene expression values were averaged for each probe over all replicates of that

### The paper explained

#### Problem
SARS-CoV-2 caused a worldwide pandemic and has already infected millions of people. Patients develop coronavirus disease 2019 (COVID-19) with variable clinical symptoms that range from mild to life-threatening. Severe COVID-19 is characterized by a strong, over-whelming immune response, and it is not clear why the antiviral response is inefficient in these cases.

#### Result
IFN-γ is a central antiviral immune mediator. We demonstrate that IFN-γ induces differentiation of epithelial colonic organoid cells, which express high levels of the SARS-CoV-2 receptor ACE2 on the cell surface. SARS-CoV-2 infection of colon organoids triggers a response that mimics IFN-γ signaling. Thus, IFN-γ signaling renders organoids susceptible to SARS-CoV-2 infection and enables high virus replication.

#### Impact
Our study suggests a mechanism by which SARS-CoV-2 exploits IFN-γ signaling in epithelial cells to increase virus production. This may explain the limited efficiency of the immune system to control infection in a subset of patients and could have implications for disease outcome and virus transmission. Balancing the IFN-γ response by pharmacological interference may offer a treatment strategy for severe COVID-19.

probe on the microarray, using the avereps function from limma. Gene set enrichment was tested either with the R package tmod or with the fgsea R package (preprint: Sergushichev et al, 2016) and with 5,000 permutations.

### Statistical analysis

Data are expressed as mean ± SEM. Differences between the different treatment groups were determined by Student's t-test or by one-way ANOVA (if more than two groups were compared) followed by Tukey's multiple comparisons test to determine differences of quantification of image analysis or expression data. $P < 0.05$ was considered significant. No statistical methods were used to predetermine sample size. All experiments were randomized and blinded whenever possible. For data visualization and statistical analysis, GraphPad Prism 8 software was used.

### Generation of t-SNE plots

Epithelial scRNAseq data from colon of healthy donors were used from a published data set (Parikh et al, 2019). Gene expression Ontology (GEO) accession number for the source data is GSE116222. Seurat R package (Butler et al, 2018) was used to perform t-distributed stochastic neighbor embedding (t-SNE) analysis. t-SNE analysis was applied to visualize cell clusters. Datasets were filtered (> 50 transcripts, < 5% mitochondrial reads per cell) and log-normalized, and the first 20 principal components were used to reduce dimensionality by t-SNE with a resolution of 0.2.

### Transmission electron microscopy

For fine structural analysis, organoids with attached Matrigel matrix were embedded in small cubes of low melting agarose. These were post-fixed with 0.5% osmium-tetroxide, contrasted with uranyl-acetate and tannic acid, dehydrated in a graded ethanol series, and infiltrated in Polybed (#02600-1, Polysciences). The cubes were placed in flat embedding molds with Polybed. After polymerization, specimens were cut at 60 nm. Specimens were analyzed in a Leo 906E transmission electron microscope (Zeiss, Oberkochen, DE) equipped with a side-mounted digital camera (Morada, SIS-Olympus, Münster, DE). Images were aligned and mounted using ImageJ (https://imagej.nih.gov/ij/).

## Data availability

Microarray data have been deposited in the Gene Expression Omnibus (GEO; https://www.ncbi.nlm.nih.gov/geo/) of the National Center for Biotechnology Information under the accession number GSE156544: https://www.ncbi.nlm.nih.gov/geo/query/acc.cgi?acc = GSE156544. All other data that support the findings of this study are available from the corresponding author on reasonable request.

**Expanded View** for this article is available online.

### Acknowledgements
The authors would like to thank Janine Wolff, Ina Wagner, Oliver Thieck, Ann Reum, and Annett Neubert for technical support. We are grateful to Rike Ziet-low for editing the manuscript. The work was supported by the DFG Grant Si-1983/3-1 and Si-1983/4-1 to MS. Open Access funding enabled and organized by Projekt DEAL.

### Author contributions
MS and JH conceived the study. RS provided surgical colon material. JH and JT prepared the materials for infection experiments, which were performed by JT and NO. DV and JT performed qPCR analyses for viral detection. CG and VB conducted the electron microscopy. ML analyzed scRNAseq data and performed experiments. JH, MS analyzed the data. H-JM performed microarray experiments and analyzed the microarray data. FT provided infrastructure support and critical scientific advice. MS provided third party funding for the study. JH and MS wrote the paper, which was read and revised by all authors.

### Conflict of interest
The authors declare that they have no conflict of interest.

### For more information
i   https://www.mdc-berlin.de/sigal
ii  https://hepatologie-gastroenterologie.charite.de/forschung/ag_gastrointe stinale_barriere_regeneration_und_karzinogenese/

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
