## [Review Process File · EMBO Molecular Medicine]

Epithelial response to IFN- γ promotes SARS-CoV-2 infection

Julian Heuberger, Jakob Trimper, Daria Vladimirova, Christian Goosmann, Manqiang Lin, Rosa Schmuck, Hans-Joachim Mollenkopf, Volker Brinkmann, Frank Tacke, Nikolaus Osterrieder, and Michael Sigal

DOI: [10.15252/emmm.202013191](https://doi.org/10.15252/emmm.202013191)

Corresponding author: Michael Sigal (michael.sigal@charite.de)

Review Timeline:

Submission Date:	20th Aug 20
Editorial Decision:	24th Sep 20
Revision Received:	22nd Dec 20
Editorial Decision:	22nd Jan 21
Revision Received:	1st Feb 21
Accepted:	2nd Feb 21

Editor: Zeljko Durdevic

Transaction Report:

24th Sep 2020

Dear Dr. Sigal,

Thank you for the submission of your manuscript to EMBO Molecular Medicine. We have now heard back from the three referees who agreed to evaluate your manuscript. As you will see from the reports below, the referees acknowledge the interest of the study. However, they raise serious and partially overlapping concerns that should be addressed in a major revision of the present manuscript.

Overall it is clear that publication of the manuscript cannot be considered at this stage. I also note that addressing the reviewers concerns in full will be necessary for further considering the manuscript in our journal and this appears to require a lot of additional work and experimentation. I am unsure whether you will be able or willing to address those and return a revised manuscript within the three to six months deadline. On the other hand, given the potential interest of the findings, I would be willing to consider a revised manuscript with the understanding that the referee concerns must be fully addressed and that acceptance of the manuscript would entail a second round of review.

Please note that EMBO Molecular Medicine encourages a single round of revision only and therefore, acceptance or rejection of the manuscript will depend on the completeness of your responses included in the next, final version of the manuscript. For this reason, and to save you from any frustrations in the end, I would strongly advise against returning an incomplete revision and would also understand your decision if you chose to rather seek rapid publication elsewhere at this stage.

I look forward to receiving your revised manuscript.

Should you find that the requested revisions are not feasible within the constraints outlined here and choose, therefore, to submit your paper elsewhere, we would welcome a message to this effect.

Yours sincerely,

Zeljko Durdevic

***** Reviewer's comments *****

Referee #1 (Comments on Novelty/Model System for Author):

Most interpretations are made on IHC stainings. And this data is mostly not quantified. Authors should try to quantify IHC and add supporting qPCR or western data that can easily be quantified. Medical Impact is there (link between IFN γ upregulation and ACE2 Expression -> suppress IFN γ Expression of epithelial cells to Change the Course of the disease), but not discussed prominently -> should be better highlighted.

Model System is very adequate, nicely demonstrating the usefulness of organoids to study infectious diseases.

Referee #1 (Remarks for Author):

Heuberger et al. report an interesting link between SARS CoV2 infection, IFN γ induction and in turn upregulation of the viral receptor ACE2. This link is mainly based on the analysis of human colon organoids, which are treated with IFN γ and also infected with SARS CoV2.

While the model system is definitely suitable to document and proof the interesting finding of manipulating the infected tissue in a way that allows even better entry for new viruses, there remain some points to be addressed.

Major points

1. Data presented in Fig. 1 is from mouse colon organoids. It is not clear to this reviewer, why the authors start out from mouse colon and then switch to human colon. Is there a specific reason for this, or can the whole manuscript be based stringently on human organoids, potentially with mouse data supporting the main findings in human tissue?
2. The authors very briefly add in the manuscript, that distal airway organoids also react on IFN γ . This is merely shown by a single qPCR bar graph. While this data is interesting, it must be supported by more data (IHC, ...) to be included. Otherwise it should be left out.
3. The whole data in Fig. 4 is nice to see. But it does not help the story. Broken down, it only shows that SARS Cov2 can infect colonic cells. This is not new and might be added as supporting material to show virus infection is possible in human colon organoids, but it does not fulfill the criteria of a full main figure in the opinion of this reviewer.
4. Discussion: There is some immediate clinical impact of the story: suppression of IFN γ response to virus infection can alter the course of the disease. Are there some hints in medical reports that support this? Please discuss this in more detail. The link you describe is a very nice finding, you should exploit it better in the discussion!

Minor points:

1. KRT20: is this truly a marker only for colon enterocytes (= colon absorptive cells). Is it not also staining other differentiated colonic cell types such as goblet cells. If so, please rename the KRT20+ population or add additional stainings for enterocytes.
2. The coexpression of Krt20 and Ace2 at the top of colon crypts is very persuasive of a co-expression in the same cell. Is there no way to do a double IHC with both markers to proof co-localization in the same cell?
3. "E-cadherin staining showed that cell polarization increased..." is E-Cad really a good marker for polarization?!?
4. "We treated organoids cultured in full medium with 100 ng/ml IFN- γ for 3 days and found that

similar to removal of Wnt, organoid proliferation decreased..." Please quantify.

5. Fig 1C: please add IHC of Ace2 on organoids.

6. Fig 1D and E: Please also show qPCR for - Wnt

7. "We treated organoids with IFN- γ for 3 days and again observed an increased abundance of KRT20 positive cells, as well as an increased thickness of the epithelial cells..." Please quantify increase of KRT20+ cells (e.g. count). Also for data in Fig. 2C. Can thickness of cells be measured by E-Cad IHC?

8. "Thus, we conclude that IFN- γ is a potent and rapid inducer of epithelial differentiation into the enterocyte lineage." Please add more markers and analyze by qPCR to outline the effect of IFN γ treatment on different cell lineages of the colon.

9. Fig. 2G. Please also add staining of colonic organoids in full medium +/- IFN γ and quantify ACE2 expression.

10. This reviewer is not sure of the data on TMPRSS2 helps the story. Better leave it out or make it more convincing.

11. This reviewer is not an expert in viral replication. But can one say that the complete viral cycle is completed by showing fusing vesicles with virus?

12. Suppl. Fig. 2b. Only brightfield is a little bit too less data. Yes, organoids do not look good, but to interpret differences into not good between the different conditions is tricky. Please quantify (e.g. FACS on apoptosis markers).

13. "... (by removal of several growth factors)": please name the growth factors here, so the interested reader does not need to search for the details in the M&M section.

14. "Thus, IFN- γ induces a differentiation program in colonic epithelial cells that is particularly apparent in "basic medium" with reduced influence of stem cell promoting factors." What does this mean, reduced influence of stem cell promoting factors? Please rephrase.

Referee #2 (Comments on Novelty/Model System for Author):

In the current study, Heuberger et al performed viral infections in organoids that were broken into cell clusters to allow viral particles to reach the apical surface of ACE2 expressing enterocytes. Since the methods for culturing human organoids as confluent monolayers on permeable supports have been established, the current study would benefit from infection studies performed in this configuration. In this platform, separate apical and basolateral compartments would allow for controlled exposure to the appropriate membrane. Furthermore, this platform will allow for more immediate measurements (e.g. polarized epithelial cytokine responses) to be made rather than waiting for 48 hours for organoids to reform following infection in broken cell clusters. In addition, the authors could also determine how basolateral exposure of organoid monolayers to IFN- γ mimics the same type of cytokine exposure that would occur in vivo.

Referee #2 (Remarks for Author):

Heuberger et al demonstrate that SARS-CoV-2 can infect human colonic enterocytes, which express ACE2. In addition, the authors demonstrate that infection can be exacerbated when concomitantly exposed to IFN- γ . The amplified degree of infection that occurs is due to increased expression of ACE2 on colonocytes as well as the increased population of Krt20+ cells that also emerges during IFN- γ exposure. The authors demonstrate that IFN- γ decreases actively dividing stem cells while increasing the number of ACE2+ colonocytes at the expense of other differentiated epithelial cell types, particularly goblet cells. The state of differentiation of human organoids dictates susceptibility to infection and response to IFN- γ .

The results of this study are important to understand the affects of SARS-CoV-2 infection on the human colonic epithelium since prior studies have evaluated infection in small intestinal epithelial cells. While the findings will contribute to our understanding of SARS-CoV-2 pathogenesis, there are some areas of concern that require clarification.

1. It is difficult to interpret the results of the study based on the methodology used to perform viral infection (please see reviewer's commentary on model system). Additional considerations include whether the epithelium in broken cell clusters is under an appropriate homeostatic state following biochemical and mechanical disruption.
2. Can the authors please explain the rationale for their media formulations or provide references? The basic media serves as the "differentiation" media; however, R-spondin, a potentiator of Wnt signaling, is present. Please confirm and/or clarify.
3. While some of the confocal images are high quality, there are several images that are overexposed. In order to make relative comparisons (since confocal is not a quantitative method), the authors should confirm that similar settings were used among all experimental conditions to obtain fluorescent intensities that are within the dynamic range of the detectors. In order to confirm changes in protein levels, other biochemical techniques such as immunoblotting are more sensitive and quantitative to support the authors' findings.
4. Did the authors confirm that all induced Krt20+ cells also express IFN-gamma receptors?
5. For EM studies, it would be useful to compare the "disintegrated" infected enterocyte to cells that are still intact within the organoid. Were lipid droplets and/or evidence of viral replication centers observed in infected human organoids?
6. Are viral particles only detected in media upon evidence of cell lysis or can new viral progeny be actively released in a polarized manner?
7. Were all human organoids generated from the same region of the colon? If so, which segment?

Referee #3 (Comments on Novelty/Model System for Author):

The experimental design relies in the use of cell models either mouse or human intestinal organoids as well as lung cells. Biological replicates are missing as organoids are being derived for one single cell line, for example. On the other hand, statistical analysis is conducted using t-Test. The novelty is medium based on the use of intestinal organoids to understand SARS-CoV-2 infection as well as the interrogation of IFN signalling in the process of viral infection by others (importantly, there are missing citations, as the recent work in Stanifer et al., 2020, Cell Reports 32, 107863). The medical impact is low with regards to the inclusion of patient sample, which could have been compensated offering a clear characterization of the cellular models used in the manuscript and the use of physiological concentrations of IFN-g. The model system is adequate, but the characterization, interpretation of the results, the lack of experiments impacts negatively in the impact of the work.

Referee #3 (Remarks for Author):

The manuscript from Heuberger and colleagues investigates how IFN- γ , a central antiviral mediator elevated in COVID19, affects epithelial cell differentiation, ACE2 expression, and susceptibility to infection with SARSCoV-2 in both colon epithelial cells and lung cells. The reason to investigate on these mechanisms is due to accumulative findings indicating the central role of SARS-CoV-2 invading epithelial cells, including those of the respiratory and gastrointestinal mucosa via ACE2 receptor.

Subsequent inflammation can promote rapid virus clearance, but severe cases of COVID-19 are

characterized by an inefficient immune response that fails to clear infection.

Heuberger and colleagues use primary epithelial organoids from human colon and observe that ACE2 is mainly expressed in the surface of enterocytes, also confirming those findings in mouse samples. Also, that inducing enterocyte differentiation in organoid culture resulted in increased ACE2. When IFN- γ treatment is used the authors observe KRT20+ enterocytes which also express high levels of ACE2. Similarly, IFN- γ promoted expression of ACE2 in human primary lung cells. IFN- γ -driven differentiation increased susceptibility to SARS-CoV-2 infection and electron microscopy revealed that the virus can efficiently complete a replication cycle in IFN- γ -treated enterocytes. Based on those observations the authors conclude that infection-induced epithelial interferon signaling promoted enterocyte maturation and enhanced ACE2 expression and suggest on the identification of a mechanism by which IFN- γ -driven inflammatory responses may increase susceptibility to SARS-CoV-2 and promote robust viral replication.

Major comments:

-Considering the physiological levels for IFN- γ found in COVID19 patients (50 pg/ml) and the concentrations used in the manuscript (100ng/ml) further experiments should be assessed to confirm on the role of IFN- γ in mouse and human organoids.

-Authors do not show protein expression levels by immunofluorescence/WB in any of the different experimental set ups. These experiments are needed to further sustain the authors hypothesis on the role of IFN- γ in SARS-CoV-2 infection.

-To unambiguously address the importance of the IFN-mediated response in enterocytes, authors may block the endogenous production of ISGs to clearly demonstrate that IFN- γ -driven inflammatory responses increases susceptibility to SARS-CoV-2 infection. This can be assessed using the pan-JAK inhibitor (pyridone-6), which inhibits the STAT1 phosphorylation activation to block the production of ISGs.

-It is well established that that IFN- γ , in synergy with TNF- α , exerts a bi-phasic effect on intestinal epithelial cell proliferation and apoptosis, by sequential modulation of the serine-threonine protein kinase AKT- β -catenin and Wnt- β -catenin signaling pathways. This has been described in a manuscript also cited in the Discussion of the manuscript (Nava et al.,) In order to further validate the use of human/mouse organoids to study the role of IFN- γ in SARS-CoV-2 infections authors should provide more evidences supporting the utility of these systems to reproduce physiological responses. In this regard, the authors should provide indications, at least of Wnt activation upon IFN- γ . By adding just results on Wnt activation and AKT the authors may provide more information on the proliferative to anti-proliferative phenotype encountered during acute intestinal inflammation upon IFN- γ exposure in vitro and in vivo.

Minor comments:

-what N fluorescence refers to should be indicated in the main text

-Figure 1: Please write protein names using capital letters. In panel A) information of the signaling pathways should be removed. If KRT20 and ACE2 are in green, also color DAPI in blue in the IF images. In qPCR plots indicate "Relative expression" and indicate then name of the gene used for it.

-Figure 2. tSNE should be changed for UMAP. Project the different genes in one single plot in different colors to highlight co-expression. Names in the Figure legend and Figure do not match (for example FM is written in the legend, but in the figures is indicated as "full"; please do revise all these issues in all the figures).

-Figure 3: please do change the nomenclature in A) panel to the used treatment (+/- IFN- γ). B-E

panels for qPCR on viral mRNA detection should be properly indicated in the y axes. F and G panels should be moved to Suppl. Figures and top ranked genes should be validated, at least, by qPCR. GO terms should be changed to "GO" terms. "disco" should be changed to "DISCO". B

-Figure 4: It is difficult to observed SARS-CoV-2 in the current semithin sections. This Figure should be repeated. Please do check other recent publications to present data supporting the word in the text section. Also, CryoEM or other techniques are currently available to provide convincing evidence of the "life cycle" of the virus.

-There are many citations which are not in the correct format. Please revise it.

Point-by-point response

Referee #1 (Comments on Novelty/Model System for Author):

Most interpretations are made on IHC stainings. And this data is mostly not quantified. Authors should try to quantify IHC and add supporting qPCR or western data that can easily be quantified.

Medical Impact is there (link between IFN γ upregulation and ACE2 Expression -> suppress IFN γ Expression of epithelial cells to Change the Course of the disease), but not discussed prominently -> should be better highlighted.

Model System is very adequate, nicely demonstrating the usefulness of organoids to study infectious diseases.

We thank the reviewers for their overall positive evaluation and have addressed the points raised in detail below

Referee #1 (Remarks for Author):

Heuberger et al. report an interesting link between SARS CoV2 infection, IFN γ induction and in turn upregulation of the viral receptor ACE2. This link is mainly based on the analysis of human colon organoids, which are treated with IFN γ and also infected with SARS CoV2.

While the model system is definitely suitable to document and proof the interesting finding of manipulating the infected tissue in a way that allows even better entry for new viruses, there remain some points to be addressed.

Major points

1. Data presented in Fig. 1 is from mouse colon organoids. It is not clear to this reviewer, why the authors start out from mouse colon and then switch to human colon. Is there a specific reason for this, or can the whole manuscript be based stringently on human organoids, potentially with mouse data supporting the main findings in human tissue?

We agree that the human data are of primary importance for this study, thus we moved the murine work to the supplement - they now can be found as Figure S1. This retains the focus on the human data and clinical relevance, while still enabling us to make the point that the mechanisms we describe are conserved between the two species.

2. The authors very briefly add in the manuscript, that distal airway organoids also react on IFN γ . This is merely shown by a single qPCR bar graph. While this data is interesting, it must be supported by more data (IHC, ...) to be included. Otherwise it should be left out.

We agree with the reviewer. Having discussed this point with the editor as well, we have now decided to follow the suggestion and remove the lung data from the manuscript.

3. The whole data in Fig. 4 is nice to see. But it does not help the story. Broken down, it only shows that SARS Cov2 can infect colonic cells. This is not new and might be added as supporting material to show virus infection is possible in human colon organoids, but it does not fulfill the criteria of a full main figure in the opinion of this reviewer.

We have now rearranged the figures and integrated the EM images in Fig. 2, as they are highly supportive of the observations presented there.

4. Discussion: There is some immediate clinical impact of the story: suppression of IFN γ response to virus infection can alter the course of the disease. Are there some hints in medical reports that support this? Please discuss this in more detail. The link you describe is a very nice finding, you should exploit it better in the discussion!

Thank you for this important suggestion. We have revised the introduction and discussion to stress the potential clinical impact of our data. For example we state in the discussion now:

"..These data suggest that infection-driven inflammation may create a vulnerable state enabling high viral replication and release, which may have consequences for the clinical course of the disease as well as for the transmission of the virus.."

As well as:

"Several clinical trials are currently investigating the role of both, activation of IFN signaling as well as inhibition of this signaling pathway via Jak-Stat inhibitors on the outcome of SARS-CoV-2 infection (Satarker, Tom et al., 2020). These studies will reveal whether in a clinical setting this signaling pathway confers beneficial immunity, or whether the virus is able to hijack this immune pathway by mechanisms that we describe here and inhibition of this pathway reduces viral replication and severity of COVID19. In addition to having a potential impact on the clinical outcome, IFN- γ may have an impact on viral transmission, as our data suggest that differentiated cells have high virus loads that are released upon cell disintegration. "

Minor points:

1. KRT20: is this truly a marker only for colon enterocytes (= colon absorptive cells). Is it not also staining other differentiated colonic cell types such as goblet cells. If so, please rename the KRT20+ population or add additional stainings for enterocytes.

Using scRNA-seq, we demonstrate that KRT20 is quite a specific marker of surface enterocytes and is only weakly expressed in a subpopulation of goblet cells in the human system (Fig. 1A). We now specify our description to "KRT20-producing enterocytes", as they can be clearly distinguished from goblet cells morphologically. Overall, our data indicate that IFN- γ promotes differentiation into enterocytes, while goblet cells are lost, which we validate using several markers of enterocytes and goblet cells, as shown in Figure 1 C-E and Suppl. Fig. 1 D.

2. The co-expression of Krt20 and Ace2 at the top of colon crypts is very persuasive of a co-expression in the same cell. Is there no way to do a double IHC with both markers to proof co-localization in the same cell?

We obtained new antibodies and now integrated double IHC for ACE2 and KRT20 on human (Fig. 1) and mouse tissue (Suppl. Fig. 2). In both, we observed ACE2 expression on the surface colonocytes.

3. "E-cadherin staining showed that cell polarization increased..." is E-Cad really a good marker for polarization?!?

Our wording might have been misleading since organoid cells are already polarized anyway. We wanted to convey that the cell shape is more columnar after IFN- γ treatment, indicating cellular differentiation. E-cad helps in this regard, as it labels cellular boundaries. Instead of referring to polarization, we now state that the cells appeared more columnar, resembling differentiated enterocytes.

To address the reviewer's concern, we now also stained the cells with phalloidin and measured the basal-apical cell length to support the differentiation after IFN- γ treatment with quantitative data (Suppl. Figs. 1 and 2). (See also reviewer's comment 7. on Fig. 2C)

4. "We treated organoids cultured in full medium with 100 ng/ml IFN- γ for 3 days and found that similar to removal of Wnt, organoid proliferation decreased..." Please quantify.

To address this concern, we performed 2 h EdU labeling and quantified the EdU positive cells. The data show that proliferation is reduced upon 3-day IFN- γ treatment, similar to removal of Wnt (Suppl. Fig. 2 E)

5. Fig 1C: please add IHC of Ace2 on organoids.

We now added staining for ACE2 on human organoids to Fig. 1J.

6. Fig 1D and E: Please also show qPCR for - Wnt

Since the reviewer suggested focussing on the human organoid system, we decided to follow this suggestion and analysed marker genes in the human system. We found that removal of growth factors induces increased expression of enterocyte and goblet cell makers (Fig. 1 E-H, Suppl. Fig. 1).

7. "We treated organoids with IFN- γ for 3 days and again observed an increased abundance of KRT20 positive cells, as well as in increased thickness of the epithelial cells..." Please quantify increase of KRT20+ cells (e.g. count).

We thank the reviewer for this suggestion and measured a) the proportion of KRT20 + cells vs. KRT20 negative cells and b) the immunofluorescence intensity of KRT20 (Suppl. Fig. 1A). The data clearly show an increase in KRT20+ cells upon differentiation, as well as an increase in the KRT20 IF signal intensity upon IFN- γ treatment.

Also for data in Fig. 2C. Can thickness of cells be measured by E-Cad IHC?

We measured the length of cells (from basal to apical site) using phalloidin staining, which supports the observation of increased cell height (along the apical to basal axis) upon IFN- γ treatment (Suppl. Fig. 1C on the right)

8. "Thus, we conclude that IFN- γ is a potent and rapid inducer of epithelial differentiation into the enterocyte lineage." Please add more markers and analyze by qPCR to outline the effect of IFN γ treatment on different cell lineages of the colon.

We thank the reviewer for that comment. We used qPCR to further analyze epithelial cell markers for enterocytes (ALPI), secretory enteroendocrine (CHGA) and goblet cells (Muc2 and ITF). Removal of growth factors induced differentiation, with increased expression of enterocyte and goblet cell markers, while IFN- γ treatment selectively promoted enterocyte differentiation, as indicated by a reduction in goblet cell marker expression (Fig. 1 E-H, Suppl Fig. 1 E). These data correlate with our IHC observations shown in Suppl. Fig. 1D.

9. Fig. 2G. Please also add staining of colonic organoids in full medium +/- IFN γ and quantify ACE2 expression.

We added staining for ACE2 for the different media conditions (see Fig. 1J).

10. This reviewer is not sure of the data on TMPRSS2 helps the story. Better leave it out or make it more convincing.

We thank the reviewer for this statement and decided to follow their suggestion to remove these data.

11. This reviewer is not an expert in viral replication. But can one say that the complete viral cycle is completed by showing fusing vesicles with virus?

We agree that the wording might be misleading. We do observe a significant increase in virus RNA from 24 h to 48 h post-infection in IFN- γ - treated cultures (now added to Fig. 2F), confirming that the virus replicated and produced new viral particles, which we can visualize using EM. We therefore conclude that virus replication occurs in these cells, and we now state it in this way in the revised version of the manuscript:

"Thus, our data indicate that SARS-CoV-2 can invade and efficiently replicate in IFN- γ treated enterocytes, which upon infection display high virus loads, promoting their disintegration and apoptosis."

12. Suppl. Fig. 2b. Only brightfield is a little bit too less data. Yes, organoids do not look good, but to interpret differences into not good between the different conditions is tricky. Please quantify (e.g. FACS on apoptosis markers).

We have now stained the SARS-CoV2 infected organoids with the apoptosis marker cleaved-caspase3, and quantified the relative abundance of apoptotic cells per organoid (Fig. 2H). The quantification confirmed that the number of apoptotic cells increases in highly infected conditions.

We would like to point out that we were not able to run FACS analyses of infected samples, due to legal restrictions with respect to working with infected samples.

13. "... (by removal of several growth factors)": please name the growth factors here, so the interested reader does not need to search for the details in the M&M section.

We thank the reviewer for this suggestion and included in the results part, in addition to the M&M, the information of the supplemented growth factors in the full medium "...in full medium (supplemented with the growth factors WNT, Noggin and EGF)..." .

14. "Thus, IFN- γ induces a differentiation program in colonic epithelial cells that is particularly apparent in "basic medium" with reduced influence of stem cell promoting factors." What does this mean, reduced influence of stem cell promoting factors? Please rephrase.

We thank the reviewer for that comment and rephrased the sentence stating now more generally: "Thus, IFN- γ rapidly induces a differentiation program in organoids towards the enterocyte lineage."

Referee #2 (Comments on Novelty/Model System for Author):

In the current study, Heuberger et al performed viral infections in organoids that were broken into cell clusters to allow viral particles to reach the apical surface of ACE2 expressing enterocytes. Since the methods for culturing human organoids as confluent monolayers on permeable supports have been established, the current study would benefit from infection studies performed in this configuration. In this platform, separate apical and basolateral compartments would allow for controlled exposure to the appropriate membrane. Furthermore, this platform will allow for more immediate measurements (e.g. polarized epithelial cytokine responses) to be made rather than waiting for 48 hours for organoids to reform following infection in broken cell clusters. In addition, the authors could also determine how basolateral exposure of organoid monolayers to IFN-gamma mimics the same type of cytokine exposure that would occur in vivo.

Referee #2 (Remarks for Author):

Heuberger et al demonstrate that SARS-CoV-2 can infect human colonic enterocytes, which express ACE2. In addition, the authors demonstrate that infection can be exacerbated when concomitantly exposed to IFN-gamma. The amplified degree of infection that occurs is due to increased expression of ACE2 on colonocytes as well as the increased population of Krt20+ cells that also emerges during IFN-gamma exposure. The authors demonstrate that IFN-gamma decreases actively dividing stem cells while increasing the number of ACE2+ colonocytes at the expense of other differentiated epithelial cell types,

particularly goblet cells. The state of differentiation of human organoids dictates susceptibility to infection and response to IFN-gamma. The results of this study are important to understand the effects of SARS-CoV-2 infection on the human colonic epithelium since prior studies have evaluated infection in small intestinal epithelial cells. While the findings will contribute to our understanding of SARS-CoV-2 pathogenesis, there are some areas of concern that require clarification.

1. It is difficult to interpret the results of the study based on the methodology used to perform viral infection (please see reviewer's commentary on model system). Additional considerations include whether the epithelium in broken cell clusters is under an appropriate homeostatic state following biochemical and mechanical disruption.

We adopted the methodology from Lamert et al. (Science, 2020), who infected small intestinal organoids following mechanical disruption. This "opening" process makes the apical epithelial surface accessible for infection. In our hands this method appeared to be the most suitable to study SARS-CoV-2 infection. We did follow the reviewer's suggestion and generated air-liquid-interphase cultures for infection experiments. We find that apical infection of ALI cultures is possible and have added this finding to the paper (Supplementary Fig. S3A). However, we also found that ALI cultures, even when grown in full medium, appear to be quite differentiated and also do not grow as pure monolayers but tend to self-assemble into organoid-like structures. Indeed, human colon organoid cultures are quite difficult to grow compared to those from other GI regions, and ALI cultures, although used by others, are less established and not well standardized. The efficient infectivity of organoids and their robust responses to growth factors and IFN- γ seem to represent major advantages for the present study. We have added these considerations also to the manuscript: "We also grew colon cells as polarized, 2-dimensional air-liquid interface (ALI) cultures, infected them from the apical side with SARS-CoV-2 confirmed infectivity in this system (Suppl. Fig. 3A). However, ALI cultures showed less robust growth and already consisted of high levels of differentiated cells in full medium conditions, overall indicating that 3D organoids are more appropriate for studying effects of differentiation and infection with SARS-CoV-2."

2. Can the authors please explain the rationale for their media formulations or provide references? The basic media serves as the "differentiation" media; however, R-spondin, a potentiator of Wnt signaling, is present. Please confirm and/or clarify.

We adopted different medium compositions and components based on previous publications (e.g. Michels et al., J Exp Med, PMID 30792186, Sato et al, Gastroenterology, PMID 21889923), and succeeded when we generated our own adaptation of the grow factor combination. We then withdrew factors step by step to achieve differentiation, which was possible with the "basic medium" we used in our paper. The presence of R-spondin alone was not sufficient to activate Wnt signaling nor did it prevent differentiation. Instead, it appears to stabilize the organoids somewhat. We now cite the most important literature on colon organoids and state that it was the basis for the development of our cultures.

3. While some of the confocal images are high quality, there are several images that are overexposed. In order to make relative comparisons (since confocal is not a quantitative method), the authors should confirm that similar settings were used among all experimental conditions to obtain fluorescent intensities that are within the dynamic range of the detectors. In order to confirm changes in protein

levels, other biochemical techniques such as immunoblotting are more sensitive and quantitative to support the authors' findings.

This is indeed an important point. Firstly, we would like to point out that all pictures from an experiment were taken using the same settings, which is the reason why some images appear overexposed. To avoid overexposure, we have now adjusted the settings to the strongest IF signal and have performed more quantitative analyses. Quantification of IF images provides some advantages: we performed quantification of the proportion of Krt20 cells, and measured the signal intensities per cell, which confirmed our qPCR results. We also included a western blot of mouse organoids showing the increase of Krt20 protein upon IFN- γ treatment. However, western blotting requires high numbers of organoids and because of the limited additional information to be gained from that, in combination with ongoing restrictions of our laboratory work due to the current pandemic, we were not able to elaborate more on this.

4. Did the authors confirm that all induced Krt20+ cells also express IFN-gamma receptors?

We analyzed single cell transcriptomic data to compare gene expression between cell types for mouse colon epithelium (Suppl. Fig. 2B) and found that IFN- γ receptors are expressed in enterocytes. We further analyzed the single cell RNA-seq data of human colon (see Fig. 1A) and show that IFNGR1 and IFNGR2 is highly and broadly expressed in Krt20+ enterocytes.

5. For EM studies, it would be useful to compare the "disintegrated" infected enterocyte to cells that are still intact within the organoid. Were lipid droplets and/or evidence of viral replication centers observed in infected human organoids?

The initial images did show viral particles in intact organoids and in a cell from a "disintegrated" organoid. Indeed, we do find high numbers of viruses in "disintegrated" cells, but after discussing this with our EM collaborators, we want to be careful about extrapolating quantitative information from EM images. Moreover, as suggested by Reviewer 1 and the editor, we de-emphasized the EM findings somewhat, moving the key findings to Fig.2.

6. Are viral particles only detected in media upon evidence of cell lysis or can new viral progeny be actively released in a polarized manner?

Our data indicate that the virus has an impact on cell survival and cells that have a high viral load appear disintegrated, which is supported by the cleaved caspase staining we have added to the revised manuscript (Fig. 2H). While this suggests that cell death could be a mechanism of virus release, it does remain possible that active release in a polarized manner also occurs. We feel that our system does not allow this question to be addressed fully and thus want to refrain from speculation on that matter.

7. Were all human organoids generated from the same region of the colon? If so, which segment?

We collected colon epithelia from surgical resection of the sigmoid colon for this study. This has now been specified in M&M.

Referee #3 (Comments on Novelty/Model System for Author):

The experimental design relies in the use of cell models either mouse or human intestinal organoids as well as lung cells. Biological replicates are missing as organoids are being derived for one single cell line, for example. On the other hand, statistical analysis is conducted using t-Test.

The novelty is medium based on the use of intestinal organoids to understand SARS-CoV-2 infection as well as the interrogation of IFN signalling in the process of viral infection by others (importantly, there are missing citations, as the recent work in Stanifer et al., 2020, Cell Reports 32, 107863). The medical impact is low with regards to the inclusion of patient sample, which could have been compensated offering a clear characterization of the cellular models used in the manuscript and the use of physiological concentrations of IFN-g. The model system is adequate, but the characterization, interpretation of the results, the lack of experiments impacts negatively in the impact of the work.

Referee #3 (Remarks for Author):

The manuscript from Heuberger and colleagues investigates how IFN- γ , a central antiviral mediator elevated in COVID19, affects epithelial cell differentiation, ACE2 expression, and susceptibility to infection with SARSCoV-2 in both colon epithelial cells and lung cells. The reason to investigate on these mechanisms is due to accumulative findings indicating the central role of SARS-CoV-2 invading epithelial cells, including those of the respiratory and gastrointestinal mucosa via ACE2 receptor.

Subsequent inflammation can promote rapid virus clearance, but severe cases of COVID-19 are characterized by an inefficient immune response that fails to clear infection.

Heuberger and colleagues use primary epithelial organoids from human colon and observe that ACE2 is mainly expressed in the surface of enterocytes, also confirming those findings in mouse samples. Also, that inducing enterocyte differentiation in organoid culture resulted in increased ACE2. When IFN- γ treatment is used the authors observe KRT20+ enterocytes which also express high levels of ACE2. Similarly, IFN- γ promoted expression of ACE2 in human primary lung cells. IFN- γ -driven differentiation increased susceptibility to SARS-CoV-2 infection and electron microscopy revealed that the virus can efficiently complete a replication cycle in IFN- γ -treated enterocytes. Based on those observations the authors conclude that infection-induced epithelial interferon signaling promoted enterocyte maturation and enhanced ACE2 expression and suggest on the identification of a mechanism by which IFN- γ -driven inflammatory responses may increase susceptibility to SARS-CoV-2 and promote robust viral replication.

Major comments:

-Considering the physiological levels for INF- γ found in COVID19 patients (50 pg/ml) and the concentrations used in the manuscript (100ng/ml) further experiments should be assessed to confirm on the role of INF- γ in mouse and human organoids.

We thank the reviewer for this critical comment and further elucidated the responsiveness of organoids to lower interferon concentrations in more detail. We found a robust response of organoid cells to IFN- γ within the range of 5 ng/ml to 100 ng/ml, while lower concentrations showed a somewhat variable response. We have included this in the manuscript as Supplementary Fig. 1B and explicitly that we chose a concentration of 50 ng/ml for our organoid experiments, which is within this range.

Since organoids are supplemented by a variety of growth factors that are likely present in supra-physiological concentrations, it is possible that higher concentrations of IFN- γ might be required to induce robust effects compared to the in vivo situation. Moreover, we would like to raise the point that local INF- γ concentrations in highly inflamed tissues are likely much higher than concentrations detected in serum.

-Authors do not show protein expression levels by immunofluorescence/WB in any of the different experimental set ups. These experiments are needed to further sustain the authors hypothesis on the role of IFN- γ in SARS-CoV-2 infection.

We have shown protein expression by immunofluorescence in several figures of the original manuscript. We have now added several new immunofluorescence analyses and have performed quantification for Krt20 expression in several settings (for example Supplementary Fig. 1A, and Fig. 3E of the revised manuscript). In mouse organoids, we could show by western blot that Krt20 levels increase upon INF- γ treatment. We noticed that relatively high quantities of proteins are required for WB, which requires high numbers of organoid cultures. Given the current restrictions of our laboratory work due to the pandemic, we were not able to further elaborate on this point, but do believe that the new results obtained from qPCR, transcriptome and immunostaining analyses provide a clear and cohesive picture of the role of INF- γ in SARS-CoV-2 infection.

-To unambiguously address the importance of the IFN-mediated response in enterocytes, authors may block the endogenous production of ISGs to clearly demonstrate that IFN- γ -driven inflammatory responses increases susceptibility to SARS-CoV-2 infection. This can be assessed using the pan-JAK inhibitor (pyridone-6), which inhibits the STAT1 phosphorylation activation to block the production of ISGs.

We thank the reviewer for this great idea. We have now combined SARS-CoV-2 infection and pyridone-6 treatment and found that pyridone-6 did indeed reduce expression of ACE2, KRT20 and the SARS-CoV-2-induced INF target genes CXCL11 and IFIT3 upon infection. Furthermore, we observed that pyridone-6 treatment reduced the virus load of IFN- γ treated infected organoids. These data are now included as Supplementary Fig. S4)

-It is well established that that IFN- γ , in synergy with TNF- α , exerts a bi-phasic effect on intestinal epithelial cell proliferation and apoptosis, by sequential modulation of the serine-threonine protein kinase AKT- β -catenin and Wnt- β -catenin signaling pathways. This has been described in a manuscript also cited in the Discussion of the manuscript (Nava et al.,) In order to further validate the use of human/mouse organoids to study the role of INF- γ in SARS-CoV-2 infections authors should provide

more evidences supporting the utility of these systems to reproduce physiological responses. In this regard, the authors should provide indications, at least of Wnt activation upon IFN- γ . By adding just results on Wnt activation and AKT the authors may provide more information on the proliferative to anti-proliferative phenotype encountered during acute intestinal inflammation upon IFN- γ exposure in vitro and in vivo.

We agree in principle that it would be very interesting to study the function of IFN- γ in more detail. There are several studies that suggest potential effects through different signaling pathways, but the exact signaling events are not fully understood. We also discussed this point with the editor and came to the conclusion that it might be beyond the scope of the study, which focuses on the effects of IFN signaling on ACE2 expression and SARS-CoV 2-infection. However, we also performed EdU incorporation after IFN- γ treatment and could recapitulate the anti-proliferative phenotype encountered during acute intestinal inflammation (Suppl. Fig. 2E).

Minor comments:

-what N fluorescence refers to should be indicated in the main text

We added this information in the main text.

-Figure 1: Please write protein names using capital letters. In panel A) information of the signaling pathways should be removed. If KRT20 and ACE2 are in green, also color DAPI in blue in the IF images. In qPCR plots indicate "Relative expression" and indicate then name of the gene used for it.

We optimized our figures and included these points.

-Figure 2. tSNE should be changed for UMAP. Project the different genes in one single plot in different colors to highlight co-expression. Names in the Figure legend and Figure do not match (for example FM is written in the legend, but in the figures is indicated as "full"; please do revise all these issues in all the figures).

We corrected the incoherent labels. We showed the tSNE plot to be consistent with the original publications. We tried to project different colors in single plots but this made the interpretation rather more difficult and we would suggest maintaining the current type of data presentation. Names have been adjusted accordingly.

-Figure 3: please do change the nomenclature in

A) panel to the used treatment (+/- IFN- γ).

We also included +/-IFN- γ to Fig.3 A.

B-E panels for qPCR on viral mRNA detection should be properly indicated in the y axes.

We changed the y-axes labels to "rel.virus genome quantity".

F and G panels should be moved to Suppl. Figures and top ranked genes should be validated, at least, by qPCR.

We have now validated the target genes CXCL11, IFIT3 and ACE2 as well as KRT20 (now added as Fig. 3F).

Go terms should be changed to "GO" terms. "disco" should be changed to "DISCO".

We have changed this accordingly.

B

-Figure 4: It is difficult to observed SARS-CoV-2 in the current semithin sections. This Figure should be repeated. Please do check other recent publications to present data supporting the word in the text section. Also, CryoEM or other techniques are currently available to provide convincing evidence of the "life cycle" of the virus.

We were not able to further elaborate on this, due to current restrictions in the labs of our collaborators. We originally did attempt to perform cryo-EM, which was not successful due to the Matrigel embedding of the cells. Since Reviewer 1 and also the editor suggested to de-emphasize the message from the EM images, we have now incorporated the most comprehensive images to Fig. 2 in order to demonstrate the presence of virus. We have also removed the confusing term "life cycle of the virus" from the text and instead added data that demonstrate an increase in virus load over time in the cultures, comparing 24 and 48 h post-infection, which suggests that viral replication does occur in these cells.

-There are many citations which are not in the correct format. Please revise it.

We thank the reviewer and corrected the citations.

22nd Jan 2021

Dear Dr. Sigal,

Thank you for the submission of your revised manuscript to EMBO Molecular Medicine. I am pleased to inform you that we will be able to accept your manuscript pending the following final amendments:

- 1) With the beginning of the new year, we encountered high number of submissions, so that our data editors were not able to process all received manuscripts. Therefore, we will send you the document with data editor's suggestions as soon as our data editors process your manuscript. Please do not submit your revised manuscript before we send you the file with data editor's suggestions. Thank you for your understanding.
- 2) Please address all concerns raised by the referees. No additional experiments are required.

***** Reviewer's comments *****

Referee #1 (Remarks for Author):

Major Points

- Answer to Major Point 1 from first review:

Authors have now moved the mouse data into a supplementary figure. Still, there is a whole paragraph on the mouse data in the results section. This reviewer still thinks that this is too much. Yes, there is a conserved mechanism between human and mouse. However, the data does not add anything to the story; no additional points can be made towards the mechanism of action of IFN γ or SARS infection of colonic tissue. In my opinion, the paragraph can be included into the human paragraph, stating at the appropriate positions that the same is seen in mouse.

- Authors show nicely, that IFN γ leads to differentiation of enterocytes, that these enterocytes express higher levels of ACE2 and that this allows more virus to infect the enterocytes. Now in the revised manuscript they give a very short glimpse on how the mechanism of SARS-CoV2 could be that leads to the observed differentiation of cells in the organoids: that indeed SARS-CoV2 induces IFN γ signaling. Nevertheless, the only data provided is that the IFN γ target genes CXCL11 and IFIT3. This part needs to be clearly expanded and the link between SARS-CoV2 infection and IFN γ signaling further evaluated.

- The interference with the Jak/Stat pathway is a good way to look at treatment possibilities resulting from the identified relevance of IFN γ signaling in this manuscript. Nevertheless, the data is quite preliminary (two organoids? No error bars? No statistics?). Is P6 also inhibiting the SARS-CoV2 infection rate of organoids without additional IFN γ treatment? This would be important; as the message is that the virus induces IFN γ itself or at least downstream signaling and thereby promotes a better soil for self re-infection. If treatment can suppress the re-infection, this would be a highlight finding. Another one would be if steroids, one of the only helpful treatments for patients with severe SARS-CoV2 infections, does also prevent re-infection by suppressing IFN γ /Differentiation of enterocytes. Additional data from these major points should make Suppl. Figure 4 a main Figure.

Minor Points

Answers to Minor Points from first review: all well answered. Thanks!

P5 "IFN- γ treatment increased KRT20 as well as ALPI expression, while the secretory cell markers were reduced" -> only in basic medium.

P5 "Analysis of single-cell RNA sequencing data showed" -> is this own data? Please specify explicitly as done above for the human single cell seq data.

P5 "Thus, in mice IFN- γ also induces differentiation into Krt20+ enterocytes that express high levels of Ace2." -> why is there no ACE2 t-sne plot? Panel A should also include and Ace2 staining after IFN γ .

P7 "Accordingly, analysis of enterocyte specific genes upon SARS-CoV-2 infection revealed that enterocyte marker genes, including ACE2 and KRT20 were also significantly increased (Fig. 3D)."

□ This might be a significant but except for AQP8 not a really relevant change in expression. Better leave this data out.

P7 "IFN- γ treatment increased the virus load of organoids infected with SARS-CoV-2, while the JAK inhibition by P6 prevented this increase (Suppl. Fig. 4B)." -> statistics?

Figure 2A: please add arrows to cells that are considered N-protein positive. Not very clear "

Figure 1E-J: labeling of x-axis not correct (no + for Basic): it is assumed that the order should be Full, Full+IFN γ , Basic, Basic+IFN γ ? Bars for significance: not clear which columns are compared.

Supp. Figure 1: in C and E the x-axis labels are not consistent. And also not consistent with Fig. 1E-J

Supp. Figure 2H: again, the labeling of the x-axis is not consistent with previous figures, here the first columns are from Basic, and the later columns are from full medium experiments. Before, it was the other way around.

Referee #2 (Remarks for Author):

Suitable for publication.

Referee #3 (Comments on Novelty/Model System for Author):

The work from Heuberger and colleagues takes advantage of human organoids as a SARS-CoV-2 infection system to address on the impact of IFN γ on epithelial cells differentiation and priming to infection. The use of organoids is not new, with different works already making use of these and other organoid model systems to interrogate these questions. Nonetheless, now the revised manuscript has better highlighted the new results and separated human vs mouse organoid data also focusing and providing quantitative data on the effect of IFN γ priming SARS-CoV-2 infection as well as differentiation in these model systems. The medical impact is medium with regards to the lack of patient data (i.e., organoids derived from elder vs young donors, in where IFN γ -related responses should be different, etc), though this reviewer understands the limitations to such studies and time constraints. However, the present manuscript now better discuss and links current observations with the importance of IFN γ in COVID19.

Referee #3 (Remarks for Author):

The clarity and interest of the manuscript for a nonspecialist is medium considering the fact that previous works have also touched similar questions/addressed similar experiments using organoids (scRNAseq, IF, EM, etc). Nonetheless the manuscript also seeks to better explain on the role of IFN γ on ACE2 expression and its relation to SARS-CoV-2 infection which is a really interesting issue to further explore using this an other organoid model systems. It should also be stressed that the authors have now addressed all the concerns raised by the referees and the clarity and focus of the work is now suitable for publication.

The authors performed the requested editorial changes.

***** Reviewer's comments *****

Referee #1 (Remarks for Author):

Major Points

- Answer to Major Point 1 from first review:

Authors have now moved the mouse data into a supplementary figure. Still, there is a whole paragraph on the mouse data in the results section. This reviewer still thinks that this is too much. Yes, there is a conserved mechanism between human and mouse. However, the data does not add anything to the story; no additional points can be made towards the mechanism

of action of IFN γ or SARS infection of colonic tissue. In my opinion, the paragraph can be included into the human paragraph, stating at the appropriate positions that the same is seen in mouse.

- Authors show nicely, that IFN γ leads to differentiation of enterocytes, that these enterocytes express higher levels of ACE2 and that this allows more virus to infect the enterocytes. Now in the revised manuscript they give a very short glimpse on how the mechanism of SARS-CoV2 could be that leads to the observed differentiation of cells in the organoids: that indeed SARS-CoV2 induces IFN γ signaling. Nevertheless, the only data provided is that the IFN γ target genes CXCL11 and IFIT3. This part needs to be clearly expanded and the link between SARS-CoV2 infection and IFN γ signaling further evaluated.

Our Microarray data show the effect of IFN γ and SARS-CoV-2 infection and the most significantly changed genes are listed in TableEV1 and TableEV2. We further validated four of the genes (ACE2, KRT20, CSCL11 and IFIT3). We agree with the reviewer that the link between IFN γ signaling and SARS-CoV-2 is of high interest, however, mechanistic in-depth studies to address this appear to be beyond the scope of this project.

- The interference with the Jak/Stat pathway is a good way to look at treatment possibilities resulting from the identified relevance of IFN γ signaling in this manuscript. Nevertheless, the data is quite preliminary (two organoids? No error bars? No statistics?). Is P6 also inhibiting the SARS-CoV2 infection rate of organoids without additional IFN γ treatment? This would be important; as the message is that the virus induces IFN γ itself or at least downstream signaling and thereby promotes a better soil for self re-infection. If treatment can suppress the re-infection, this would be a highlight finding. Another one would be if steroids, one of the only helpful treatments for patients with severe SARS-CoV2 infections, does also prevent re-infection by suppressing IFN γ /Differentiation of enterocytes. Additional data from these major points should make Suppl. Figure 4 a main Figure.

We could show that interference with Jak/Stat signaling reduced the virus production in IFN γ treated organoids, which indicates that reducing epithelial IFN γ responses limits virus production.

We would like to clarify that we performed these experiments in organoids derived from two patients, not two organoids. To avoid misunderstanding, we now clarified this point and changed the label to patient 1 and patient 2. Since we could only perform experiments on limited numbers, we did not perform any statistical test. Of course, it would be desirable to study the effect on IFN γ interference in more detail, but we were currently unable to expand on this due to the limitations associated with the pandemic itself.

Minor Points

Answers to Minor Points from first review: all well answered. Thanks!

P5 " IFN- γ treatment increased KRT20 as well as ALPI expression, while the secretory cell markers were reduced" -> only in basic medium.

We added “only in basic medium” to clarify this finding.

P5 "Analysis of single-cell RNA sequencing data showed" -> is this own data? Please specify explicitly as done above for the human single cell seq data.

We added “....Single-cell RNA sequencing data from a recently published data set accessible by an online tool” ...

P5 "Thus, in mice IFN- γ also induces differentiation into Krt20+ enterocytes that express high levels of Ace2.." -> why is there no ACE2 t-sne plot? Panel A should also include and Ace2 staining after IFNg.

In this single cell data set Ace2 expression appears very low and therefore the dataset is not suitable to present Ace2 expression, which is likely due to limitations of single cell sequencing depth. However, co-expression is shown using immunofluorescence in the same figure.

P7 "Accordingly, analysis of enterocyte specific genes upon SARS-CoV-2 infection revealed that enterocyte marker genes, including ACE2 and KRT20 were also significantly increased (Fig. 3D)."

\ This might be a significant but except for AQP8 not a really relevant change in expression. Better leave this data out.

Since we find various enterocyte-specific genes to be significantly upregulated in our transcriptome data sets we believe it is important to show these data. We also validated the results using qRT-PCR for key genes. Therefore, we suggest to keep the data in the figure, but can of course remove them upon editorial request.

P7 "IFN- γ treatment increased the virus load of organoids infected with SARS-CoV-2, while the JAK inhibition by P6 prevented this increase (Suppl. Fig. 4B)." -> statistics?

We were only able to perform experiments on a limited number of samples, and therefore did not perform any statistical tests. Of course, it would be desirable to study the effect on INFG interference in more detail, which we currently could not expand due to the limitations associated with the pandemic itself. However, we think that these first results should be shown, as they could be useful to stimulate further research into this direction.

Figure 2A: please add arrows to cells that are considered N-protein positive.

We have added arrows to Figure 2A.

Not very clear " Figure 1E-J: labeling of x-axis not correct (no + for Basic): it is assumed that the order should be Full, Full+IFNg, Basic, Basic+IFNg? Bars for significance: not clear which columns are compared.

We thank the reviewer for carefully evaluating the figures. Indeed, the “+” for “basic” had shifted to the “Full” line. We also optimized the bars for significance and added lines to show which columns are compared.

Supp. Figure 1: in C and E the x-axis labels are not consistent. And also not consistent with Fig. 1E-J

We thank the reviewer for carefully evaluating the figures and now changed the x-axis labels, which are consistent with the Figure 1 E-J now.

Supp. Figure 2H: again, the labeling of the x-axis is not consistent with previous figures, here the first columns are from Basic, and the later columns are from full medium experiments. Before, it was the other way around.

Again, we thank the reviewer for carefully evaluating the figures. We have now brought the columns into a consistent order and adjusted the x-axis for consistency with the other figures.

Referee #2 (Remarks for Author):

Suitable for publication.

We thank the reviewer for the positive evaluation.

Referee #3 (Comments on Novelty/Model System for Author):

The work from Heuberger and colleagues takes advantage of human organoids as a SARS-CoV-2 infection system to address on the impact of IFN γ on epithelial cells differentiation and priming to infection. The use of organoids is not new, with different works already making use of these and other organoid model systems to interrogate these questions. Nonetheless, now the revised manuscript has better highlighted the new results and separated human vs mouse organoid data also focusing and providing quantitative data on the effect of IFN γ priming SARS-CoV-2 infection as well as differentiation in these model systems. The medical impact is medium with regards to the lack of patient data (i.e., organoids derived from elder vs young donors, in where IFN γ -related responses should be different, etc), though this reviewer understands the limitations to such studies and time constraints. However, the present manuscript now better discuss and links current observations with the importance of IFN γ in COVID19.

Referee #3 (Remarks for Author):

The clarity and interest of the manuscript for a nonspecialist is medium considering the fact that previous works have also touched similar questions/addressed similar experiments using organoids (scRNAseq, IF, EM, etc). Nonetheless the manuscript also seeks to better explain on the role of IFN γ on ACE2 expression and its relation to SARS-CoV-2 infection which is a really interesting issue to further explore using this an other organoid model systems. It should also be stressed that the authors have now addressed all the concerns raised by the referees and the clarity and focus of the work is now suitable for publication.

We thank the reviewer for the positive evaluation.

2nd Feb 2021

Dear Dr. Sigal,

We are pleased to inform you that your manuscript is accepted for publication.

Corresponding Author Name: Michael Sigal

Journal Submitted to: Embo Molecular Medicine

Manuscript Number: EMM-2020-13191